# A gradient-boosted tree framework to model the ice thickness of the World's glaciers (IceBoost v1.1)

Niccolò Maffezzoli[1,2,3], Eric Rignot[2,4], Carlo Barbante[1,3], Troels Petersen[5], and Sebastiano Vascon[1]

[1]Ca' Foscari University of Venice, Venezia, Italy
[2]University of California Irvine, Irvine, USA
[3]Institute of Polar Sciences, National Research Council, Venezia, Italy
[4]Jet Propulsion Laboratory, Pasadena, USA
[5]Niels Bohr Institute, University of Copenhagen, Copenhagen, Denmark

**Correspondence:** Niccolò Maffezzoli (niccolo.maffezzoli@unive.it)

**Abstract.** Knowledge of glacier ice volumes is crucial for constraining future sea level potential, evaluating freshwater resources, and assessing impacts on societies, from regional to global. Motivated by the disparity in existing ice volume estimates, we present IceBoost, a global machine learning framework trained to predict ice thickness at arbitrary coordinates, thereby enabling the generation of spatially distributed thickness maps for individual glaciers. IceBoost is an ensemble of two gradient-boosted trees trained with 3.7 million globally-available ice thickness measurements and an array of 39 numerical features. The model error is similar to existing models outside polar regions and up to 30-40% lower at high latitudes. Providing supervision by exposing the model to available glacier thickness measurements reduces the error by up to a factor 2 to 3. A feature ranking analysis reveals that geodetic information are the most informative variables, while ice velocity can improve the model performance by 6% at high latitudes. A major feature of IceBoost is a capability to generalize outside the training domain, i.e. producing meaningful ice thickness maps in all regions of the World, including in the ice sheet peripheries.

## 1 Introduction

Under atmospheric heating by human forcing, with few exceptions, glaciers have been retreating worldwide at unprecedented rates (Hugonnet et al., 2021), with projections predicting one third of the mass loss at the end of the century in the most optimal $+1.5°C$ scenario (Rounce et al., 2023). At present, glacier melting contributes to sea level rise equally to ice sheets (Zemp et al., 2019; Masson-Delmotte et al., 2021), with far reaching implications for coastal communities worldwide (Pörtner et al., 2019). Ice mass loss from glacier shrinkage also impacts water availability for an estimated population of 1.9 billion people living in or depending on glacier-sourced freshwater (Huss and Hock, 2018; Rodell et al., 2018; Immerzeel et al., 2020).

Accurate and continuous knowledge of glacier ice thickness spatial distributions over time is thus of critical importance to inform and refine numerical models to better simulate future scenarios under a fast-changing climate (Zekollari et al., 2022). Measurement campaigns and surveys have led to direct ice thickness measurements for about 3,000 of the existing more than 216,000 glaciers (Welty et al., 2020). The data is unsurprisingly sparse, albeit radar surveys from airborne campaigns have

significantly increased the amount of measurements and coverage, particularly over polar regions. Knowledge of absolute glacier volume thus heavily relies on models, or physical and mathematical interpolations.

An array of models has been proposed over time, with varying degrees of applicability (Farinotti et al., 2017). Only two have been applied to all glaciers on Earth. They are based on principles of ice flow dynamics and use surface characteristics, including ice surface velocity. The mass conservation approach by Huss and Farinotti (2012) has been extended with four additional regional models to produce a global consensus ensemble (Farinotti et al., 2019). More recently, Millan et al. (2022) also provided a global solution, leveraging a complete coverage of glacier velocities and using a shallow-ice approximation (Cuffey and Paterson, 2010) with basal sliding. The degree of applicability of thickness inversion methods is broad. The main challenges relate to the validity of the underlying physical assumptions, parameter calibration, and availability and quality of model input data. As an example, the shallow ice approximation simplifies the stress field by neglecting longitudinal stresses, lateral drag, and vertical stress gradients. This setup works well in the interior of ice sheet, or with ice masses with small depth to width ratios. On mountain glaciers, Le Meur et al. 2004 found that this approach can be acceptable for slope values less than 20%. In relation to challenges of parameter calibration, Millan et al. 2022 tuned the creep parameter A on regional-basis, using ice thickness measurements, if present, and, if not, from other regional averaged values. The basal sliding was parametrized indirectly by imposing assumptions of surface ice velocity-to-slope ratios. Input data are crucial - for example, a shallow ice approximation can only by used if the ice velocity product is available. For glaciers not covered by this data, the method cannot be applied. Previous methods have also suffered from incomplete surface ice velocity, which is now becoming a globally available, as well as quality surface mass balance data. We refer the readers to Farinotti et al. 2017 for a comprehensive overview of thickness inversion models, their advantages and limitations.

A parallel line of research is exploring machine learning methods. Few approaches based solely on deep learning have been explored so far. Clarke et al. (2009) proposed a multilayer perceptron trained on neighboring deglaciated regions to reconstruct glacier bedrocks. This method does not invoke physics but assumes similarity between glacier bedrocks and topography of nearby ice-free landscapes. Convolutional neural networks (CNN) are now the state-of-the-art architectures for physical models emulators, and they have gained traction in glaciology with Jouvet et al. (2022); Jouvet (2023). Trained to represent physical models with much cheaper computation cost, emulators have the versatility to both compute forward modeling and to invert for ice thickness. Uroz et al. (2024) trained a CNN to produce ice thickness maps on 1,400 Swiss glaciers, by ingesting surface velocity and Digital Elevation Model (DEM) maps, with their ground truth consisting of ice thickness fields obtained by a combination of experimental data and glaciological modeling. Growing attention is being directed to physics-informed neural networks, as they provide a natural setup to both generate an approximate solution of a differential equation and minimize the misfit with observational data, if any. For a review, we refer the readers to Liu et al. (2024).

In this work, we present IceBoost, a machine learning system designed for modeling ice thickness across all of Earth's glaciers, including continental glaciers, ice caps, and ice masses at the edges of ice sheets. The method is not explicitly based on any physical law. It is fully data-driven, yet contains the versatility to incorporate many relevant features, that do not easily lend themselves to classical physics modeling. The only theoretical consideration of significance relates to the the architecture choice. We approach the problem as a machine learning regression task, predicting ice thickness at any arbitrary point within a

glacier's boundary. IceBoost employs an ensemble of two gradient-boosted decision tree models, XGBoost (Chen and Guestrin, 2016) and CatBoost (Prokhorenkova et al., 2018), both trained with ice thickness as target. The target data is naturally tabular-structured (localized-in-space measurements) and is extracted from the Global Ice Thickness Database (GlaThiDa, or GTD hereafter), a centralized effort by the World Glacier Monitoring Service (WGMS), detailed by Welty et al. (2020). The model is informed using a set of 39 numerical features, extracted from an array of (either gridded or tabular) products and organized in a tabular structure. While CNN are best suited to operate on gridded products (images), when data is tabular-structured, tree-based models often provide a much faster and powerful alternative to neural networks, as highlighted from theoretical considerations (Grinsztajn et al., 2022) and empirically from machine learning practitioners. CNNs are also typically very demanding in terms of training data, computation power, and require elaborate methods for the interpretability of internal layers. Instead, gradient-boosted systems allows for a straightforward interpretability in terms of which features are important for the thickness inversion of individual glaciers, using Shapley values (Lundberg and Lee, 2017)

In the following sections we introduce the model concepts (Sect. 2), describe feature interpretability (Sect. 3), illustrate the model inference and compare its performance against existing global solutions (Sect. 4), describe some limitations and possible improvements (Sect. 5), consider the computational cost (Sect. 6), before we conclude (Sect. 7).

## 2    Methods

Hereafter, we describe the datasets used to generate the features for model training and inference. The mass balance feature is presented in Appendix A for brevity.

### 2.1    Datasets

As ground truth, we use the GlaThiDa v. 3.1.0 dataset (GlaThiDa Consortium 2020; Welty et al. 2020), which comprises 3.8 million ice thickness measurements, integrated with an additional 11,000 measurements from 44 glaciers included in the IceBridge MCoRDS L2 Ice Thickness product (Paden et al., 2010). The model is trained using the glacier geometries digitized and stored in the Randolph Glacier Inventory (RGI) version v.62 (Pfeffer et al., 2014). This version extends version 6 with additional 1,000 ice bodies from the Greenland periphery (connectivity level 2 to the ice sheet), hereafter still referred to as glaciers. RGI v.62 thus provides the opportunity to train and test the model ability to reproduce thickness patterns in an ice sheet flow domain, a region with an extensive amount of available thickness data from the IceBridge mission. RGI glacier geometries include both the glacier external boundaries and the ice-free regions contained therein (hereafter referred to as nunataks). At inference time, IceBoost supports geometries from either RGI v.62 (n=216,502) or RGI v.70 (n=274,531).

The features used to train the model are computed from various datasets, presented hereafter (Table 1). Elevation and geodetic information are computed from the global Tandem-X 30m Edited Digital Elevation Model (EDEM, Bueso-Bello et al. 2021; González et al. 2020; Martone et al. 2018). Modeled spatially distributed surface mass balance from Greenland and Antarctica is obtained from the Regional Atmospheric Climate Model (RACMO2) products, available at different spatial resolutions (Noël et al. 2018; Noël and van Kampenhout 2019; Noël et al. 2023). Glacier-integrated mass balance values are

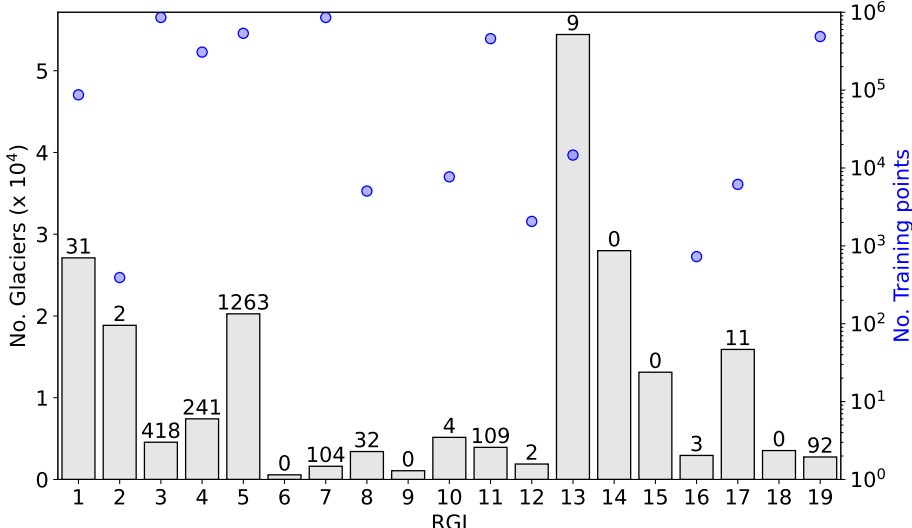

**Figure 1.** Statistics of glaciers and training data for each Randolph Glacier Inventory (RGI) region. The total number of glaciers in each region is represented by bar lengths on the left axis (in units of $10^4$). The numbers over each bar represent the absolute counts of glaciers with available training data (blue circles, right axis). Out of 216,000 glaciers worldwide, only 2,300 contain at least one training point. RGI regions 6, 9, 14, 15 and 18 have no training data.

imported from Hugonnet et al. (2021). Temperature at 2-meter (t2m, hereafter) fields are taken from ERA5 and ERA5-Land
(Hersbach et al., 2020; Muñoz-Sabater et al., 2021). Surface ice velocity fields are taken from Millan et al. (2022), except for glaciers listed in Antarctica (RGI 19) and Greenland (RGI 5), where we use the velocity products from Mouginot et al. 2019 and Joughin et al. 2016, respectively.

  Our solutions are compared with existing global models, while acknowledging that additional solutions exist with regional validity. Outside the ice sheets, we compare with Farinotti et al. (2019)'s ensemble and with the shallow ice approximation
by Millan et al. (2022). In the Antarctic periphery (RGI 19), we compare with Farinotti et al. (2019) and BedMachine v3 (Morlighem, 2022a). In the Greenland periphery, we compare with Farinotti et al. (2019) and BedMachine v5 Morlighem (2022b). BedMachine Greenland and Antarctica are complete modeled bed topographies and ice thickness maps of the two ice sheets. They are constructed using different methods, in different regions, and continuously updated. Mass conservation is typically applied where the knowledge of surface speed is robust and sufficient. Elsewhere, other methods like kriging
intepolation and streamline diffusion are adopted. At the time of writing, BedMachine Greenland v5 uses Millan et al. (2022) for isolated glaciers and ice caps, and mass conservation or kriging interpolation elsewhere.

| Feature | Variable name | Local | Unit | No. variables | Primary dataset | Horizontal resolution |
|---|---|---|---|---|---|---|
| Curvature | $c_{50}, c_{100}, c_{150},$ $c_{300}, c_{450}, c_{gfa}$ | ● | $0.01\text{m}^{-1}$ | 6 | Tandem-X EDEM | 30 m |
| Distance from no ice | $d_{noice}$ | ● | km | 1 | RGI | 30 m |
| Distance from ocean | $d_{ocean}$ | ● | km | 1 | RGI, GSHHG | 200 m |
| Elevation | $z$ | ● | m | 1 | Tandem-X EDEM | 30 m |
| Elevation normalized | $z_{01}$ | ● | m | 1 | Tandem-X EDEM | 30 m |
| Elevation above base | $z - z_{min}$ | ● | m | 1 | Tandem-X EDEM | 30 m |
| Glacier cluster area | $A_{cluster}$ | | $\text{km}^2$ | 1 | RGI | 30 m |
| Glacier area | $Area$ | | $\text{km}^2$ | 1 | RGI | 30 m |
| Glacier aspect | $Aspect$ | | degrees | 1 | Tandem-X EDEM | 30 m |
| Glacier curvature | $Curvature$ | | $0.01\text{m}^{-1}$ | 1 | Tandem-X EDEM | 30 m |
| Glacier elevation delta | $\Delta z$ | | m | 1 | Tandem-X EDEM | 30 m |
| Glacier length | $L_{max}$ | | m | 1 | RGI | 30 m |
| Glacier mass balance (geodetic) | $MB$ | | m w.e.yr$^{-1}$ | 1 | Hugonnet et al. (2021) | - |
| Glacier perimeter | $Perimeter$ | | m | 1 | RGI | 30 m |
| Glacier slope | $Slope$ | | - | 1 | Tandem-X EDEM | 30 m |
| Glacier min, max, median elevation | $z_{min}, z_{max}, z_{med}$ | | m | 3 | Tandem-X EDEM | 30 m |
| Mass balance | $mb$ | ● | m w.e.yr$^{-1}$ | 1 | Hugonnet et al. (2021) RACMO2 | 30 m 1-2 km |
| Slope | $s_{50}, s_{75}, s_{100}, s_{125},$ $s_{150}, s_{300}, s_{450}, s_{gfa}$ | ● | - | 8 | Tandem-X EDEM | 30 m |
| Temperature at 2 meters | $t2m$ | ● | K | 1 | ERA5, ERA5-Land | 31 km, 9 km |
| Velocity | $v_{50}, v_{100}, v_{150},$ $v_{300}, v_{450}, v_{gfa}$ | ● | m yr$^{-1}$ | 6 | Millan et al. (2022) Joughin et al. (2016) Mouginot et al. (2019) | 50 m 250 m 450 m |

**Table 1.** List of features and products used by IceBoost, alongside their units, number of variables, and primary dataset for their calculation. Local features are flagged by circles. No circles indicate glacier mean values. RGI shapefiles are mostly derived from 30 m resolution satellite data, therefore variables calculated from RGI are indicated with 30 m horizontal resolution. The Tandem-X EDEM has a 3 arc-second horizontal resolution, corresponding to ca. 30 meters at the equator. The model is trained with ice thickness data from the GlaThiDa Consortium (GlaThiDa Consortium, 2020). See Figure 2 for an analysis of the predictive power of the features.

## 2.2 Training features

The model is trained with a set of 39 numerical features, extracted from the above-mentioned datasets (Table 1). Some features are local, i.e., vary within the glacier; others are per-glacier constants. We use the main variables used for a steady-state mass conservation-based inversion or physical-based approximations (Huss and Farinotti, 2012; Millan et al., 2022): ice velocity, mass balance and spatial first and second-order gradients of the elevation field (hereafter referred to as slope and curvature, respectively). We extend the amount of information with geometrical features that relate to topographies (e.g. U-shaped valleys) crafted in alpine landscapes (distance to margins or internal deglaciated patches), and metrics of glacier size which are typical of scaling approaches (Bahr et al., 2015). We also use variables that carry a fingerprint of thermal regime and climate setting (temperature and distance from the ocean). It is worth mentioning that in a gradient-boosted tree approach, the same variable can be used more than once in the decision tree scheme. For training, the features are calculated and stored offline in a training dataset. At inference time, all features are calculated on-the-fly.

### 2.2.1 Constant features

The following features are per-glacier constants: area ($Area$), perimeter ($Perimeter$), glacier minimum ($z_{min}$), maximum ($z_{max}$) and median ($z_{med}$) elevation, glacier elevation delta ($\Delta z = z_{max} - z_{min}$), length ($L_{max}$), area of glacier cluster ($A_{cluster}$).

The elevations are calculated from the DEM. The areas, perimeter and glacier length are calculated from the glacier geometries. The areas only include the iced region and exclude nunataks. The area of a glacier cluster is determined by summing the areas of all glaciers connected to the glacier under investigation, considering connections up to a maximum level of 3. This feature indicates whether a glacier is isolated ($Area = A_{cluster}$) or part of a larger system. The glacier length ($L_{max}$) is calculated as the maximum distance between any pair of points that lie on the glacier convex hull (Appendix A1). The glacier convex hull is referred to as the smallest convex set that contains the glacier itself. Except for the elevations, the other constant features are metrics of glacier size. Extensive previous research has been directed to empirical scaling laws to relate glacier volume to, for example, its area (Bahr et al., 2015). IceBoost retains the information carried by metrics of glacier size in a machine-learning approach, without imposing explicit laws.

### 2.2.2 Distance from ice free regions ($d_{noice}$) and distance from the ocean ($d_{ocean}$)

Given a point $x_0$ inside a glacier $g$, we calculate the distance to the closest free pixel. Such a target point may lie within or outside the glacier. We define a glacier cluster as the collection of all neighboring glaciers. For example, three glaciers $\{g_1, g_2, g_3\}$ form a cluster if $g_1$ shares a pixel with $g_2$ and $g_2$ shares a pixel with $g_3$, despite $g_1$ and $g_3$ not being necessarily adjacent glaciers. The glacier cluster is calculated by detecting, starting from the glacier that contains the point $x_0$, all its proximal neighbors. The procedure is repeated iteratively for every neighboring glacier until no further neighbors are found. In discrete mathematics, this structure is a graph. Once the cluster is computed, all internal shared borders (the ice divides) of the cluster are removed, while internal ice-free nunataks are kept. This procedure potentially results in collections of up to

thousands of geometries per cluster. The calculations to create the cluster are made by setting up a graph-network structure,
using the NetworkX python library (Hagberg et al., 2008).

The minimum distance from the point $x_0$ to an ice-free region ($d_{noice}$) is the minimum of distance between $x_0$ and all points $x$ lying on the cluster's valid geometries:

$$d_{\text{noice}}(\mathbf{x_0}) = \arg \min_{\mathbf{x} \in \text{cluster}} d(\mathbf{x_0}, \mathbf{x}) \tag{1}$$

The valid geometries can either be the cluster's external boundaries or all the cluster's nunataks. The distances are computed by querying K-dimensional trees, an approximate nearest neighbor lookup method, on the geometries defined in the Universal Transverse Mercator (UTM) projection. We compare the proximal points obtained from this method with those from a brute-force calculation and find indiscernible results. The pipeline is computed both as a feature for the creation of the training dataset and at inference time for every generated point. For computational speedup, at inference time, the number of geometries K used by the KD-tree can be capped to the closest 10,000 geometries. An example for $d_{noince}$ is shown in Figure A1.

Similar to the distance to ice-free regions, we calculate the closest distance to the ocean. We use the Global Self-consistent, Hierarchical, High-resolution Geography Database (GSHHG), containing all the world's shoreline geometries, in resolution "f" (full). Like $d_{noice}$, $d_{ocean}$ is calculated by querying a KD-tree on the geometries, in UTM projection. We find this feature to be relatively unimportant on continental glaciers far from the coasts, but increasingly informative at high latitudes where many marine-terminating glaciers are located.

## 2.3 Elevation, curvature, slope

We use the DEM to calculate the following features: local elevation $z$, elevation above the glacier lowest elevation $z - z_{min}$, elevation normalized between 0 and 1 $z_{01} = (z - z_{min})/(z_{max} - z_{min})$, curvature and slope.

To calculate the slopes, the DEM tiles are first projected in UTM, differentiated and the resulting vector magnitude is convoluted using Gaussian filters of different kernel widths to capture the variability across different spatial scales: 50m, 75m, 100m, 125m, 150m, 300m, 450m and an adaptive filter $af$, Eq. 2:

$$af = \frac{2A}{\pi L_{max}} \tag{2}$$

where $A$ and $L_{max}$ are the area and glacier maximum length features. $af$ corresponds to the semi-minor axis of an ellipse of area $A$, and semi-major axis $L_{max}/2$. This kernels aims at estimating the glacier spatial size. For values lower than 100 or above 2000 meters, $af$ is capped to these values. Each training point yields eight slope features. The purpose of using many kernels is to allow the model the freedom to account for different glacier spatial scales. For small glaciers the small kernels are found to be more important than the bigger kernels, and vice versa (Fig. 2).

To limit the computational cost, for the calculation of the curvature, the elevation field is smoothed using only the 50, 100, 150, 300, 450 and $af$ kernels, thus resulting in 6 features per point. All geodetic features are obtained from linear interpolation.

The model is also informed with glacier-integrated (i.e. mean) values of slope, curvature, and aspect ($Slope$, $Curvature$ and $Aspect$, in Table 1).

### 2.3.1 Mass balance

Glacier thickness is controlled by ice flow and mass balance (Cuffey and Paterson, 2010). The latter is the net result of positive precipitation and mass loss mechanisms both at the surface and at the glacier bed. We use two different mass balance features. The first is the glacier-mean 2000-2020 mass balance values imported as-is from Hugonnet et al. (2021). Such dataset is available for RGI v6, and is therefore suitable to train our model and for inference over RGI v6. For inference on the latest RGI v7 geometries, we reference the two RGI datasets, or impute the regional mean for missing ids. The second mass balance feature is local map of mass balance, obtained by downscaling glacier-integrated values for non-polar glaciers, or directly interpolating RACMO for ice sheet glaciers. It is discussed in Appendix A2, and Appendix D.

### 2.3.2 Surface velocity

Presently, the surface ice velocity is available with almost global coverage, and is therefore used as input feature. The velocity magnitude fields are smoothed with the six kernels: 50m, 100m, 150m, 300m, 450m and $af$, and linearly interpolated. The velocity products used have different resolution: Millan et al. (2022) (all regions except for Greenland and Antarctica), Joughin et al. (2016) (Greenland) and Mouginot et al. (2019) have resolutions of 50m, 250 and 450 meters, respectively. If the product resolution is higher than any kernel size, the kernels are set to match the product resolution. For every training point a total of six velocity features are obtained. At inference time, the missing velocity features are treated according to the imputation policy described in Appendix B2.

### 2.3.3 Temperature

The model is informed with local 2-meter temperature (t2m), as a loose regional indicator of thermal regime and ice thickness. Although a weak indicator, this variable may still be useful in a decision tree, helping to split the data at earlier stages and enabling more powerful features to drive predictions at deeper levels of the tree structure. We also use this feature to provide regional context to a global model. We use ERA-5 Land (0.1 degree grid spacing, Muñoz-Sabater et al. 2021) and, for the missing pixels caused by imperfect fractional land masks along the coastlines and islands, we incorporate the ERA5 t2m field (0.25 degree resolution, Hersbach et al. 2020), bilinearly interpolated to the ERA-5 0.1 degree resolution. We consider monthly maps over the 2000-2010 and average them over this time period to generate one single global temperature field. This map is linearly interpolated the at the measurement points (training) or at the generated points (inference time).

### 2.4 Data pre-processing and time tagging

A significant number of zero-thickness measurements are found in GlaThiDa. While some of these measurements are found close to glacier boundaries, at times they are found inside glacier geometries. We decided to discard all GlaThiDa zero-thickness entries. GlaThiDa thickness data from glacier 'RGI60-19.01406' (peripheral glacier in Antarctica, 65.5 S, 100.8 E, maximum elevation 500 m a.s.l.) is erroneously registered with a factor 10 too much (up to more than 3,000 meters of ice). We divide this data by 10. All datasets used in this work are tied to different time intervals. The glacier outlines refer to 2000-2010

for most glaciers (Pfeffer et al., 2014). The ice surface velocity outside the ice sheets is tied to 2017-2018 (Millan et al., 2022). Tandem-X EDEM results from acquisitions between 2011 and 2015. The GlaThiDa dataset stores ice thickness measurements from 1936 up to 2017. The ERA5 and ERA5-Land fields are tagged to 2000-2010. To homogenize temporally as much as possible all datasets in the creation of the training set, while maximizing its size, all ice thickness measurements older than 2005 are discarded. In addition, we discard all measurements that lie outside glacier boundaries or inside nunataks. Overall, the model is conservatively estimated to be trained on data spanning from 2005 to 2017.

The resulting GlaThiDa dataset comprises 3.7 million points collected from 2300 glaciers (Fig. 1). To reduce the amount of correlated data close to each other and reduce computational costs, the training dataset is spatially downscaled. Each glacier is divided into a grid of 100x100 lat-lon pixels, and the per-pixel average is computed for all features and thickness data. The original 3.7 million point dataset is thus encoded into a final training dataset of 300,000 entries. For baseline comparison, the thickness fields from Millan et al. (2022) and Farinotti et al. (2019) were also downscaled.

## 2.5 Model

We utilize two Gradient-Boosting decision Tree (GBDT) models (Friedman, 2001). A GBDT model consists of multiple additive decision trees and is trained iteratively. In each iteration, a new decision tree is added and tasked to fit the residuals of the previous tree by minimizing an objective function. Training continues until a stopping criterion is met, either reaching a maximum number of iterations or detecting overfitting through a separate validation dataset. IceBoost is an ensemble model comprising two GBDT variants: XGBoost (Chen and Guestrin, 2016) and CatBoost (Prokhorenkova et al., 2018). Both models use a second order Taylor approximation of the objective function and employ a depth-wise tree growth scheme. However, CatBoost builds symmetric trees, which tends to act as a regularizer against overfitting, and handles categorical features natively without requiring one-hot encoding. We train both models independently using a squared loss, $l = (h - \hat{h})^2$, where $h$ represents the target thickness data from GlaThiDa and $\hat{h}$ represents the modeled thickness. The IceBoost ensemble combines them by averaging their respective predictions.

Despite the different climates and glacier ice flow regimes in various regions, we decide not to specialize IceBoost regionally but rather to build one single model and optimize its hyperparameters globally. The decision is driven by the ease of deployment and the availability of certain features (particularly mass balance, temperature and distance from the ocean) that can provide some regional context to the model. It should be noted that the model optimal parameters may reflect the imbalance of the training data among different regions, potentially making it slightly more biased towards polar regions where more training data is available. Potential solutions to specializing the model regionally would include optimizing the hyperparameters separately for each region and/or applying a heavier penalty within the regions of interest. We note that high thickness values are underrepresented in the training dataset (Supp. Info, Fig. 1). However, no significant bias is observed in regions with the thickest ice, such as Greenland and Antarctica (Table 2).

| RGI | Region | GTD - IceBoost | RMSE IceBoost w/ supervision | RMSE IceBoost w/o supervision | RMSE Millan et al. 2022[§] | RMSE Farinotti et al. 2019 |
|---|---|---|---|---|---|---|
| 1 | Alaska | 16 (20) | 47 (2) | 116 (21) | 151 | 173 |
| 2 | Western Canada and US | - | - | - | - | - |
| 3 | Arctic Canada North | -5 (4) | 32 (1) | 83 (7) | 131 | 129 |
| 4 | Arctic Canada South | 0 (5) | 18 (1) | 58 (9) | 103 | 115 |
| 5 | Greenland Periphery | -5 (4) | 28 (2) | 93 (23) | 112 | 112[*] |
| 6 | Iceland | - | - | - | - | - |
| 7 | Svalbard | -8 (9) | 14 (1) | 52 (7) | 66 | 51 |
| 8 | Scandinavia | -1 (9) | 20 (1) | 42 (6) | 60 | 53 |
| 9 | Russian Arctic | - | - | - | - | - |
| 10 | North Asia | -8 (5) | 4 (1) | 15 (3) | 19 | 23 |
| 11 | Central Europe | -7 (5) | 10 (1) | 35 (5) | 46 | 35 |
| 12 | Caucasus and Middle East | 16 (1) | 9 (1) | 56 (1) | 65 | 56 |
| 13 | Central Asia | -6 (6) | 8 (1) | 36 (12) | 62 | 37 |
| 14 | South Asia West | - | - | - | - | - |
| 15 | South Asia East | - | - | - | - | - |
| 16 | Low Latitudes | - | - | - | - | - |
| 17 | Southern Andes | -22 (16) | 12 (1) | 43 (8) | 35 | 40 |
| 18 | New Zealand | - | - | - | - | - |
| 19 | Antarctic and Subantarctic | 4 (10) | 47 (2) | 109 (20) | 113 | 192[*] |

**Table 2.** IceBoost performance and comparison with existing models, as measured by RMSE, on a regional basis. All units are in meters. The numbers in parentheses refer to 1 standard deviation across n=100 regional cross-validation runs. GTD: GlaThiDa.

* The evaluation is limited to glaciers with no connectivity to the ice sheet.

§ In RGI 5 and RGI 19, this column refers to BedMachine v.5 (RGI 5, Morlighem 2022b) and BedMachine v.3 (RGI 19, Morlighem 2022a).

## 2.6 Model training, hyperparameter optimization and performance

Hyperparameter tuning is conducted independently for both XGBoost and CatBoost, both referred to as "model" for simplicity, and in an identical manner, using a Bayesian hyperparameter optimization framework (*Optuna*, Akiba et al. 2019). The best parameters are determined by training each model over n=200 trials. In each trial, a different set of hyperparameters is selected, the model is trained on an 80% random split of the data, and the objective error loss is evaluated and monitored on the remaining 20% split. Considering that the target ice thickness variable is not uniformly distributed (Supp. Info, Fig. 1), to reduce the risk of overfitting to a particular data split, the 80%-20% random split is recreated every time the optimizer calls the objective function during hyperparameter search. This approach effectively incorporates the dataset's variability as part of the noise. Its main advantage is that it helps identify hyperparameters that generalize well across different data splits, aiming

for a more robust model. The trade-off is that randomizing the data splits in each trial introduces variability in the objective function, which can make convergence to the optimal hyperparameters more challenging. However, such variability in the loss is generally manageable within Bayesian optimization frameworks. To further prevent overfitting by reducing the model complexity, we enforce early stopping in each optimization trial (for both XGBoost and CatBoost). Early stopping is a form of regularization that halts training if performance on the 20% split does not improve for a fixed number of rounds, n=50. The best hyperparameters are identified as those selected in the trial for which the objective loss is minimized (Appendix A3). The XGBoost model is also optimized with respect to the following parameters that combat overfitting: $lambda$, $alpha$, and $subsample$.

We acknowledge that hyperparameter optimization is typically performed by leaving out a smaller third set for offline evaluation. However, we found that such evaluations were highly dependent on the data split, due to the stochasticity of the random split and the non-uniformity of the target variable. This dependency made the results less informative. For example, if Antarctic high-thickness data were included in the test set, the test set error would be disproportionately high. Consequently, no separate test set was created, and the entire dataset was allocated to training (80%) and validation (20%).

IceBoost performance is quantified by fixing the best set of hyperparameters, training the model and evaluating its performance regionally, using a cross-validation scheme (Table 2). We evaluate i) the median of the residual distribution $res = GTD - IceBoost$ and ii) the root mean squared-error (RMSE), on a test set consisting of a 20% random split of the regional data. Cross-validation involves training the model n=100 times, each time randomizing the regional 20% test split. Two different routines are considered to produce the 20% test split. In the first routine ('with supervision'), the 20% measurements are taken from glaciers where other data is considered for the 80% training split. This approach allows the model to be trained on one glacier data and tested on other locations within the same glacier (no data used for training is ever used for testing). In the second routine ('without supervision'), we impose a stricter constraint by creating the 20% test from completely unseen glaciers. The model performance is reported in Table 2 for regions with sufficient data in the training set (Fig. 1). We cannot evaluate the model performance in regions 2, 6, 9, 14, 15, 16, and 18 due to too few or absent data. For regions 10, 12, 13, and 17, where the limited data is available, statistics are provided but considered only indicative of regional performance. Nevertheless, a similar model behavior is likely expected for regions that are geographically close, have a similar ice flow regime and similar mean thickness or feature values: 13-14-15 (extensive Himalayan glaciers), and 6-7-9 (high latitude glaciers and ice caps), and 8-11-12-18 (small-to-medium size mountain glaciers). An example of one iteration of training without supervision is displayed in the Supplementary Information (Supp. Info., Figs. 3-4).

The performance of XGBoost and CatBoost, evaluated individually on ground truth data, is comparable within 1-sigma statistical fluctuations, with neither consistently outperforming the other (Supp. Info., Table 1). A qualitative comparison at inference time on selected glaciers suggests the same conclusion holds, with similar predicted patterns even in the absence of ground truth data (Academy of Sciences ice cap shown in Supp. Info., Fig. 2). We create IceBoost by taking an unweighted mean of XGBoost and CatBoost. Alternative approaches, such as applying regional weighting or per-glacier weighting based on feature explainability, could be explored but are left for future work.

Evaluated against ground truth data, IceBoost error is comparable to state-of-the-art global solutions outside polar regions and up to 30-40% lower in polar regions (Table 2). The much lower errors when training with supervision indicate that providing the model with glacier context proves to be beneficial. While this conclusion seems consistent on a regional scale, we find that on a glacier-by-glacier basis, the model is not always sensitive to additional tie points, regardless of where the context is provided (further discussion in Sect. 4). In the Greenland and Antarctic peripheries, it is noteworthy that Farinotti et al. (2019) model performance is only evaluated on glaciers not connected to the ice sheet. In Greenland (RGI 5), the reported statistics combine Millan et al. (2022) shallow ice approximation for glaciers and ice caps not connected to the ice sheet, with Morlighem 2022b kriging/mass conservation elsewhere. In the Antarctic periphery (RGI 19), as ground truth data are only found in the Antarctic continent (and none in the Subantarctic islands), the reported statistics refers entirely to BedMachine v.3 (Morlighem, 2022a).

## 3 Model interpretability

To understand the relative strengths of the features for the model prediction, we carry out a feature ranking analysis using SHapley Additive exPlanations (SHAP, Lundberg and Lee 2017). For the analysis we consider the XGBoost model. Shapely is a framework based on cooperative game theory where the goal is to equitably distribute the total gains to players (i.e. the model features) based on their individual contributions. A feature SHAP value reflects its marginal contribution to the model, specifically the change in the model's prediction when the feature is added or removed. Positive (negative) SHAP values indicate that the feature increases (decreases) the model prediction with respect to its average baseline (the sum of all SHAP values for a given instance equals the model's prediction for that instance), while SHAP absolute values represent the magnitude of the feature contribution to the model prediction, regardless of the direction.

A SHAP analysis is shown in Figure 2 for a random global subset of n=2,000 training data points. Each instance is represented by a dot. The features are ordered from top to bottom by decreasing mean absolute values, i.e. more important features are on top (a less informative but more compact visualization is shown in Figure A3). The feature SHAP values are the x-coordinates, while the feature values are represented in the color bar. As an example, points with high distance-from-ice-free regions values typically have higher SHAP values, i.e., will push the model towards higher thickness predictions. For almost all features except for the slope and the curvature, higher feature values will lead to higher ice thickness predictions.

Local slopes and curvature are important features, highlighting the DEM quality as a crucial input for accurate glacier thickness estimates (see Appendix B5 for DEM-driven model artifacts). The elevation from the glacier minimum, rather than absolute elevation above sea level, is found to be most informative.

The closest distance to ice-free areas is a powerful feature that often mirrors ice thickness spatial distribution in continental valley glaciers. This feature retains its power even in large glacier systems with multiple nunataks (e.g., Fig. A1).

As known from area-volume scaling models (Bahr et al., 2015), metrics for glacier extent ($Area$, $L_{max}$) are found to be powerful predictors. By providing regional context for different flow regimes and metrics of continentality, the 2-meter temperature, $t2m$, and distance to the ocean $d_{ocean}$, are found to be moderately informative. The local mass balance $mb$ (Appendix

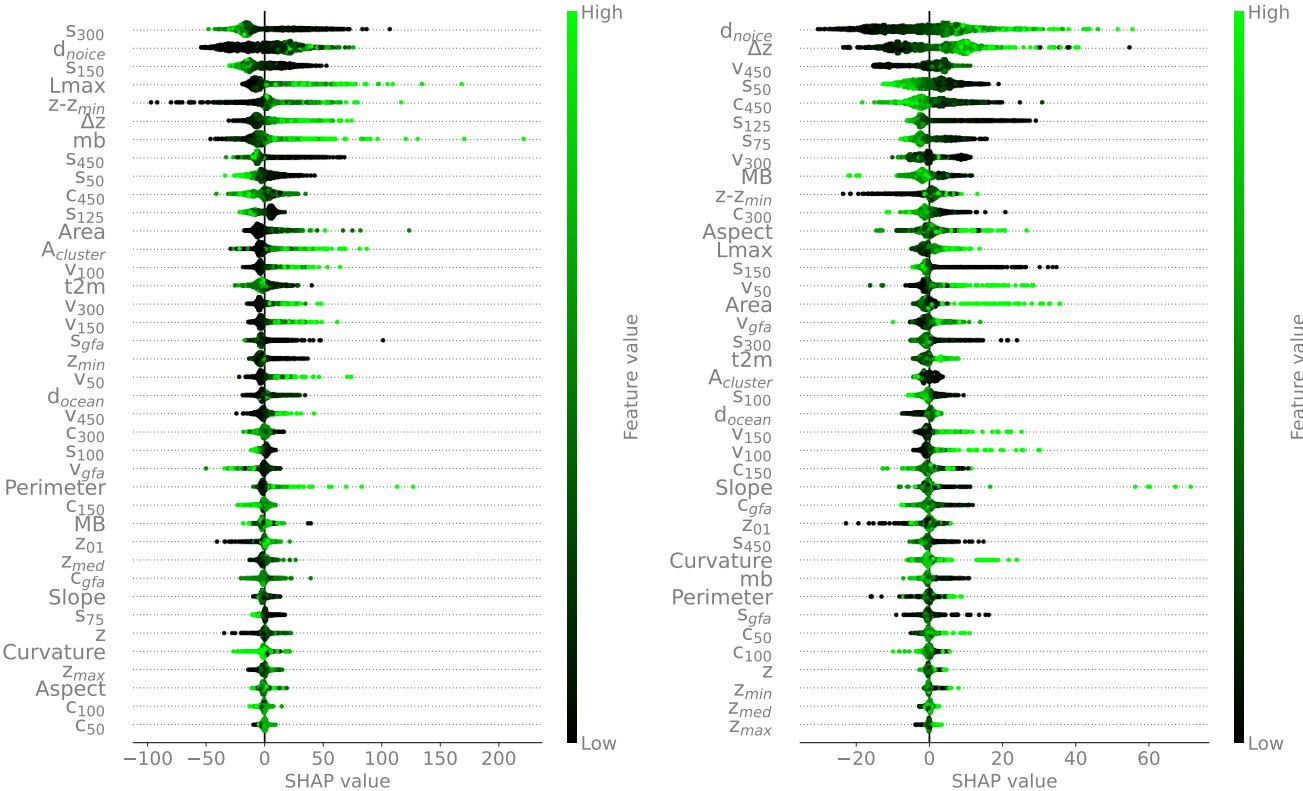

**Figure 2.** Left: SHAP analysis of n=2,000 random instances (each ice measurement instance is represented by a dot on each feature row). Features are ordered from top to bottom by decreasing mean absolute SHAP values: top features are more important. The horizontal coordinate indicates how the model output changes with respect to its baseline, in a positive or negative direction, hence how predictive are the features. The color bar reflects the normalized feature variability range. See Table 1 for the variable names. Right: the same analysis is carried out on a random set of points in RGI 11 (Central Europe). See Fig. A3 for the feature rankings based solely on absolute mean SHAP values.

A2), is also found to be relatively informative, despite our simple elevation-based downscaling mass balance approach. We

also acknowledge that most glaciers are currently out of equilibrium, likely resulting in the accumulation and ablation zones being altered by the climate signal. Ice velocity is found to be a major predictor but, perhaps surprisingly, not as strong as those mentioned above globally. Possibly, the wide range of variability across over three orders of magnitude in velocity makes this information difficult to account for, in addition, possibly, to data uncertainty. The role of surface velocity is further investigated by training the model without velocity information. We find that the error increases up to 6% maximum for high-latitude re-

gions, while no substantial difference is found elsewhere. We speculate that at high latitudes, where more extensive glaciers are located, geodetic information becomes relatively less informative (low and uniform values of slope and curvature, absolute elevations of glaciers less informative), increasing the ranking of surface velocity. Since the largest ice volumes are stored at high latitudes, all velocity features are retained in the model.

Except for the metrics related to glacier size and the glacier elevation difference $\Delta z$, all other glacier-integrated features

($Slope$, $Aspect$, $Curvature$) are found to be relatively unimportant, including glacier geodetic mass balance values, $MB$ (see also Appendix C). Overall, the analysis highlights the crucial importance of high-quality DEMs.

The analysis provides a general overview of the predicting power of the feature set by accounting for a random global set of training entries. A slight reshuffling of the feature ranking is expected, however, if evaluating glaciers individually (an example is discussed in Sect. 4) or regionally (e.g., RGI 11 in Fig. 2). The SHAP analysis proves very instructive to access the

320 information gain provided to the model by each feature. It can be used to decide which features should be retained and which ones can be dropped without substantial loss of performance.

## 4 Model deploy

At deploy time, the model ensemble is tasked to produce a continuous glacier ice thickness solution. The pipeline consists in generating $n$ discrete points randomly inside the glacier boundary and outside nunataks, calculating the feature vector $x_n$ and

325 querying the model locally $h_n = IceBoost(x_n)$. The feature vector $x_n$ is calculated on-the-fly (Appendix, B1). The glacier volume is calculated by Monte Carlo (Appendix, B3). An approximately continuous solution can be obtained in the limit $\lim_{n\to\infty} h_n(x_n)$. Typically $n = 10,000$ provides a good representation even for relatively big glaciers.

To investigate the effect of added supervision, we consider the Malaspina glacier (RGI60-01.13696). The glacier, located in coastal southern Alaska, is the world's largest piedmont glacier with an area of $3900\ \mathrm{km}^2$. Its piedmont lobe is largely grounded

below sea level. Measurements on this glacier are found in our training dataset. A recent campaign has vastly increased the amount of measurements on the glacier and provided a detailed overview of the terminus thickness distribution and bedrock topography (see Fig. 5 in Tober et al. 2023).

We train the model with and without the available measurements included in the training dataset (hereafter referred to as "with and without supervision", respectively). We note that, contrary to a kriging technique, IceBoost does not use the data

explicitly, but rather adjusts its parameters at training time. The model trained without supervision predicts an ice thickness of up to 700-800 $\mathrm{m}$ at the terminus and in other deepest parts of the glacier. Next, we include the measurements in the training set and train the model with supervision. The model output changes drastically at the terminus, with the solution values closer to the ground truth, although the model still struggles to fully capture the high thickness values that correspond to localized deep subglacial channels found by radar surveys (Tober et al., 2023). Note that the solution changes in other parts of the terminus as

well and also relatively far from the data. Training IceBoost with supervision greatly improves the model skill, suggesting that a significant advantage compared to existing approaches is achieved when data is available by: i) improving the general model performance by increasing the training data; and ii) improving the prediction on individual glaciers. While it is not trivial to understand why IceBoost prediction without supervision deviates from the ground truth for the Malaspina glacier, the model error is consistent with what has been found in this region at cross-validation (RGI 1, Table 2). The RMSE evaluated on ground

truth data (n=588) for this glacier is 147 m when the model is trained without supervision. Incorporating supervision with 20% of the data reduces the RMSE to 31 m on the remaining 80%, representing a 4.7-fold improvement. This experiment also shows

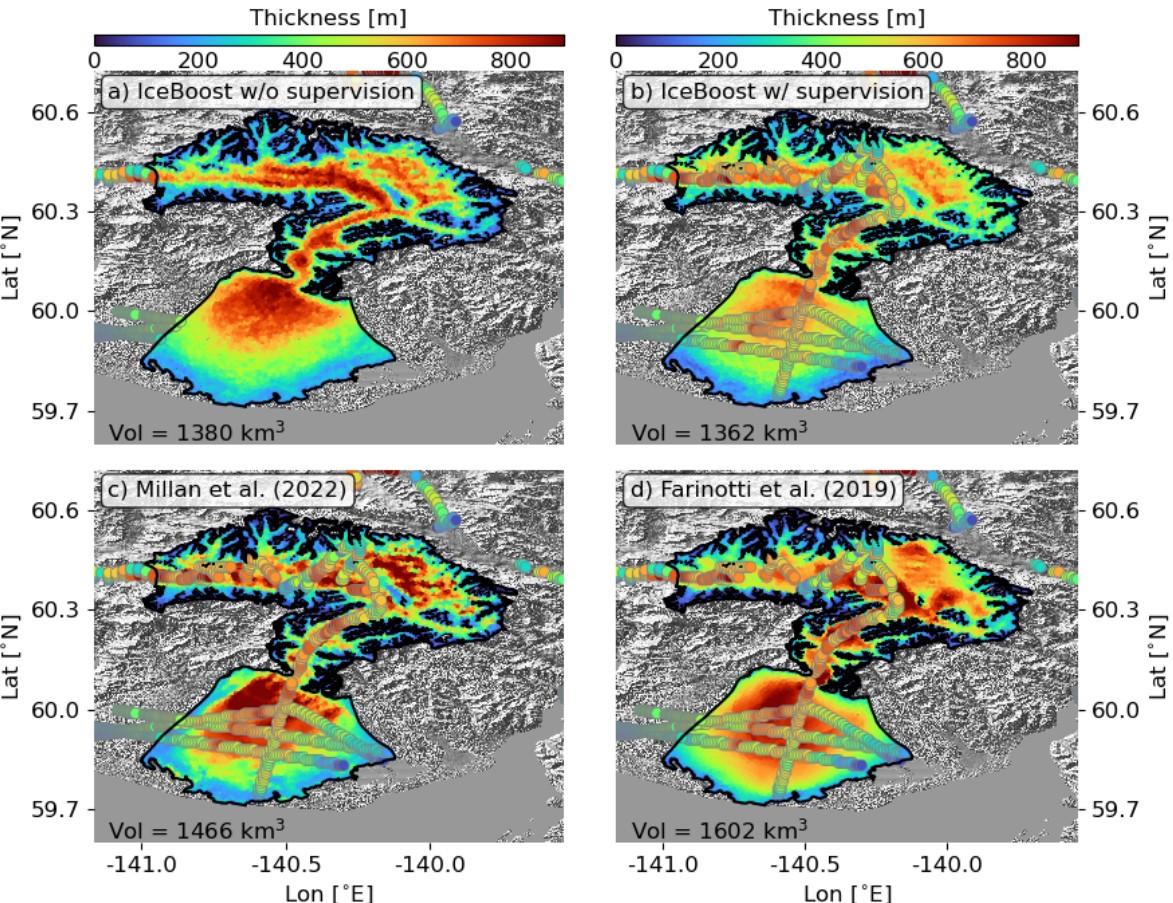

**Figure 3.** Modeling of the Malaspina glacier (RGI60-01.13696, Alaska) by IceBoost (this work, panels a and b), Millan et al. 2022 (panel c) and Farinotti et al. 2019 (panel d). IceBoost is trained without and with supervision, respectively in panels a) and b). The glacier ice volume difference between the two cases is 1.3%. In panels b), c), and d), the ground truth thickness data are represented as large circles. The Tandem-X EDEM hillshade hillshade is shown in transparency.

that, although the model does not explicitly account for dependence between points (opposite to a neural network structure), it produces a meaningful covariant pattern in both cases.

The same analysis is carried out for Mittie Glacier, a large and surge-type glacier in Arctic Canada North (Fig. 4). The RMSE is 68 m when training without supervision, and 27 m with supervision, a reduction by a factor of 2.5 - approximately half the improvement observed for the Malaspina glacier. The comparison would suggest that not all modeled glaciers equally benefit from added supervision. We speculate that the improvement in model performance obtained with supervision is likely attributable to the inclusion of high-thickness data (and/or out-of-distribution feature values), which provide valuable information for modeling deep glaciers.

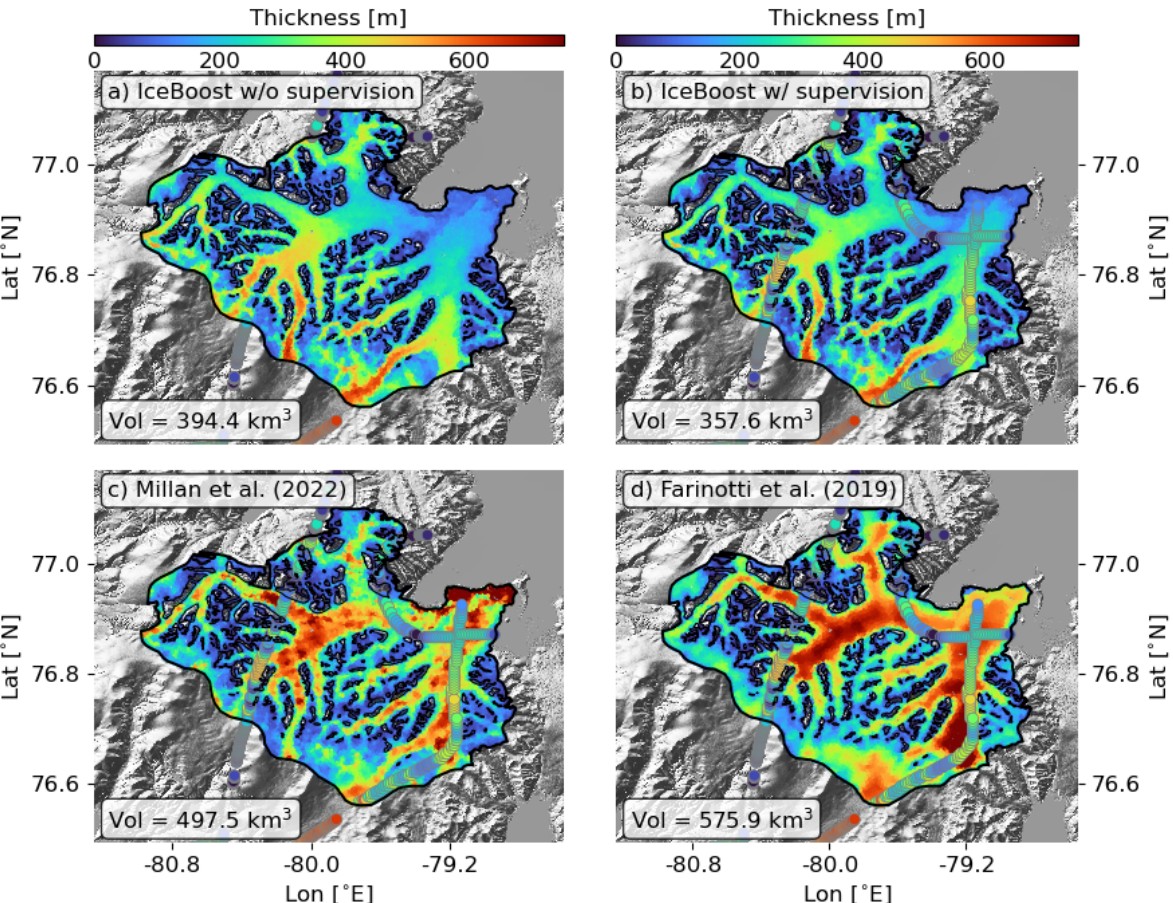

**Figure 4.** Modeling of Mittie Glacier (RGI60-03.01517, Arctic Canada N.) by IceBoost (this work, panels a and b), Millan et al. 2022 (panel c) and Farinotti et al. 2019 (panel d). IceBoost is trained without and with supervision, respectively in panels a) and b). The glacier ice volume difference between the two cases is 9.5%. In panels b), c), and d), the ground truth thickness data are represented as large circles. The Tandem-X EDEM hillshade hillshade is shown in transparency.

To assess how the feature set influences the model prediction, we conducted a SHAP analysis of the Mittie Glacier, evaluating feature SHAP values using the XGBoost module (Figs. 5, 6). At each point, the sum of all feature SHAP values, $\sum_f |SHAP(f)|$, reflect the overall contribution of the feature set to the model prediction (Fig 5a). On the Mittie Glacier, the feature set provides the most information in the deep ice rivers of the south and at the marine terminus in the northeast. Difference between XGBoost and CatBoost individual predictions reach up to 150-200 meters in the deepest regions (Fig 5b). These differences mimics the IceBoost averaged output (Fig 4b). Averaged across the whole glacier, the five most important features (inset in Fig. 5a) are the slope ($s_{300}$, $s_{150}$, $s_{450}$), the elevation above the glacier base ($z - z_{min}$), and the distance to ice-free regions ($d_{noice}$). The spatial variability in SHAP values reveals the areas where each feature contributes most (Fig. 6). The

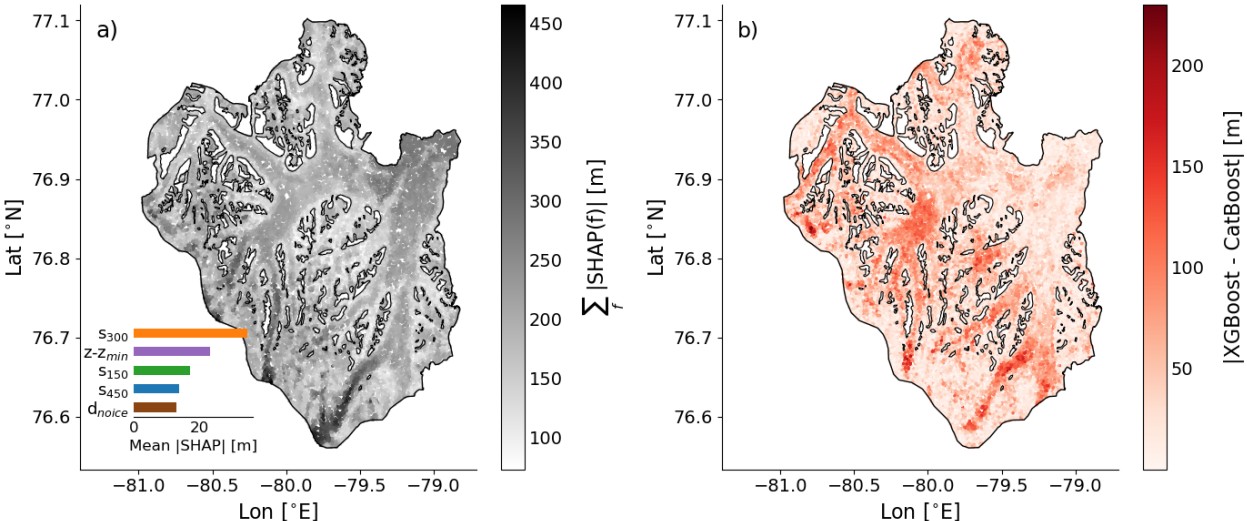

**Figure 5.** a) Sum of absolute SHAP values of all features, calculated for the Mittie glacier. Higher (lower) values indicate greater (weaker) contribution from the feature set to the model. The top-5 features are displayed in the inset, ranked from the most important to the least important by decreasing mean SHAP values. b) Absolute difference between the XGBoost and CatBoost individual modules. The reader is referred to Fig. 4 for the geographical setting of the glacier.

slope field smoothed with the widest kernel ($s_{450}$) is particularly informative near broad, flat glacier streams, while the slope smoothed with the smallest kernel ($s_{150}$) provides value over the steep, mountainous terrain located in between nunataks. The $d_{noice}$ feature is also most informative near nunataks, whereas the elevation above the base ($z - z_{min}$) is especially valuable at the marine terminus, and less so elsewhere. While the IceBoost model does not inherently provide an uncertainty estimate on predicted ice thickness, valuable insights can be derived using the Shapely analysis on the feature set, and to some extent by comparing the separate predictions of XGBoost and CatBoost.

The spatial resolution of the IceBoost model is considered hereafter. The input features are extracted from products with varying resolution (Table 1). For example, the satellite products range from 30 m (DEM) up to 250 m for surface velocity fields over the ice sheets. Convolution with various kernels of different size are also implemented when generating the features, enlarging the receptive field. Other features are per-glacier constants. The minimal spatial variation of the thickness maps generated by IceBoost, loosely referred to as the model resolution, is evaluated by visually assessing the predictions (examples in Figures 7), and is estimated to be $\simeq$100 m. The model has neither the capabilities (it is not trained to) nor the resolution to predict smaller-scale basal features, unless their fingerprints are clearly reflected on the surface.

We note that the model predicts at times rather fine-grained details in ice thickness, such as close to ice shelf transitions or in the proximity of high elevation gradients (Fig. 7).

An extensive comparison between IceBoost and other models can be found in the Supplementary Information (see Data Availability), for 190 glaciers. For each one of the 19 RGI regions, n=10 glaciers are modeled with IceBoost, and compared to

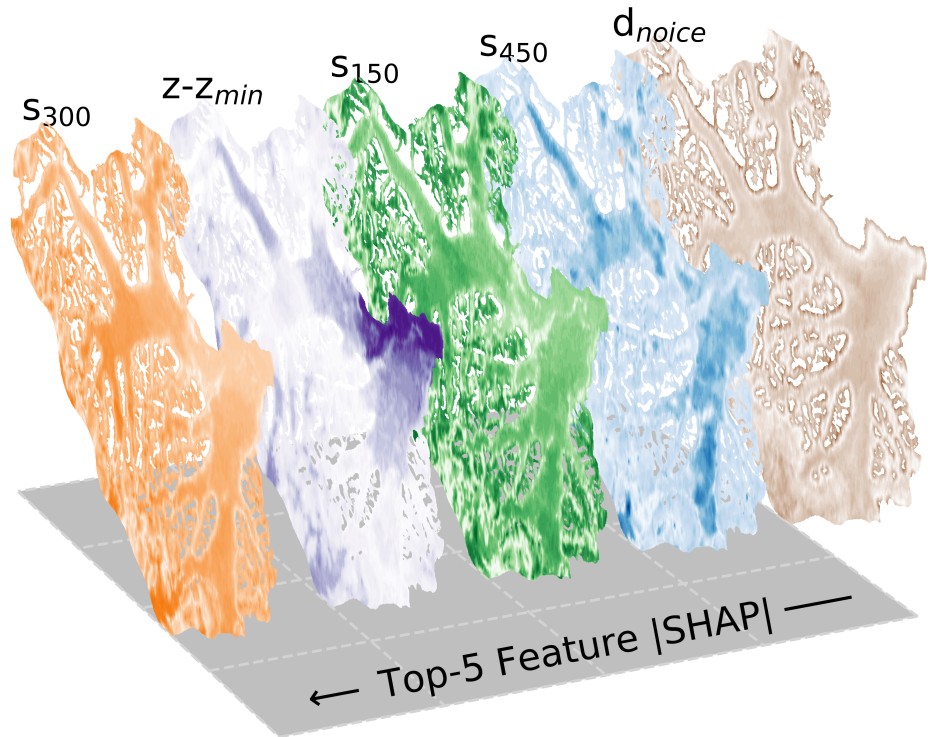

**Figure 6.** Absolute SHAP spatial variability of the top-5 ranked features, calculated for the Mittie glacier. The colors range from white (low values) to saturated colors (high values). $d_{noice}$ is important close to the glacier ice free regions. The slope features $s_{150}, s_{300}, s_{450}$ show similar patters, with the widest 450m-slope more informative on the broad ice rivers, and less valuable elsewhere. The $z - z_{min}$ feature is very informative close to the marine terminus.

Millan et al. 2022 or BedMachine (whenever applicable, RGI 5: Morlighem 2022b, RGI 19: Morlighem 2022a) and with the ensemble by Farinotti et al. 2019.

## 5 Applications, improvements, and limitations

The ice thickness maps produced by IceBoost have significant potential in numerical modeling of future glacier evolution, by providing a modern initial condition. Valuable insights can also be gained by utilizing IceBoost-derived maps, compared to
outputs from other inversion techniques, such as those by (Millan et al., 2022) and (Farinotti et al., 2019). A notable advantage of a machine learning model operating on tabular data is its flexibility to predict point estimates of other variables, such as surface ice velocity or mass balance, by setting them as training target. In these cases, ice thickness, if known, can serve as additional input feature. Ice velocity maps (or mass balance) can be generated from scratch or used to fill gaps in existing products. Additionally, modeling the ice thickness of past glaciers can be explored, if their geometry is known, by leveraging
knowledge of past feature records, or by using present-day features under specific assumptions.

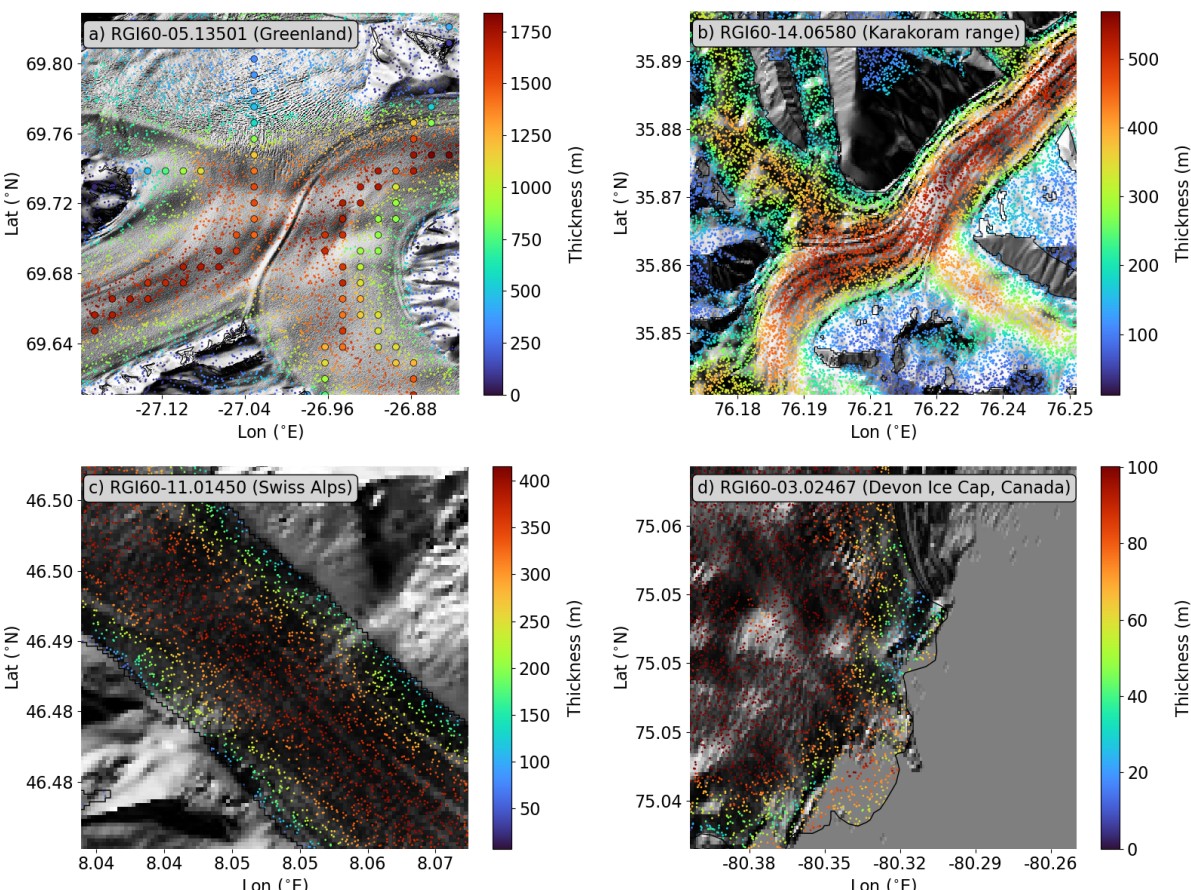

**Figure 7.** Iceboost-modeled glaciers at different spatial scales, overlayed to the Tandem-X EDEM hillshade. The small circles reflect the modeled ice thickness, evaluated at random locations. In panel a) the ground truth thickness data are represented as large circles.

Improvements to IceBoost can be explored by expanding the training dataset, both in the number of entries and the inclusion of more informative features. Prioritizing an increase in high-thickness data would be particularly beneficial. Ensembling with other machine learning models, such as a multi-layer perceptron, may further enhance performance by reducing point estimate error and improving the smoothness of the output solution, an important factor for numerical modeling. Additionally,

implementing or running in parallel other machine learning schemes that generate probability distributions as output (e.g. NGBoost (Duan et al., 2020)) could yield valuable uncertainty maps alongside predictions.

Enhancing the quality of the products used to generate the features, particularly the DEM, will certainly improve the model. Artifacts in the input DEM are the primary source of error in the generated thickness maps. They are directly reflected in the model outputs, leading to visible inaccuracies in the proximity of the artifact (Fig. B3). Artifacts can also occur when the

400 smoothing kernels applied to the input products are not appropriately sized. For instance, in the case of glacier RGI60-19.00134 (Alexander Island, Antarctica, Fig. B3), the kernels used in this study may be too small to adequately capture and remove the

elevation roughness present at the spatial scales of this glacier ($\sim 4000 \text{ km}^2$). This mismatch results in artifacts manifesting as (likely unrealistic) high ice thickness gradients (Fig. B3).

## 6  Computational cost

The memory load for creating the training dataset is 80 Gbytes, primarily due to memory necessary to import, merge and operate on the DEM tiles. Downgrading to Tandem-X 90m would certainly reduce the computational cost, at the expense of accuracy. Model training and deploy can be done on either GPU or CPU. Model training requires a few minutes. The inference phase requires generating the feature vector on-the-fly and querying the model. The former requires between 1 second to 1 minute for the most complex glaciers, almost independently of the choice of number of points to generate ($10^4 - 10^5$). The latter is faster and requires ca. $10^{-1}$ s/glacier. The feature generation dominates the computational cost, and parallelization with multiple processors is implemented in the model for regional simulations. For higher spatial details, the point density can be selectively increased locally up to $O(10^5)$. We recommend not increasing $n$ above a million points as the information gain is limited by the model resolution ($\simeq 100 \text{ m}$). Hard disk memory recommendations are 10-500 Gbytes. All of Earth's glaciers can be conservatively run on 1 Tbytes hard disk, 128 Gbytes RAM and 1-20 CPUs. Unless feature calculation is moved to GPU (for e.g. see RAPIDS Development Team 2023), a graphics card is deemed unnecessary since time overhead of data transferring surpasses the benefit of a marginally faster model query on GPU.

## 7  Conclusions

To the best of our knowledge, IceBoost is the first machine-learning model capable of predicting ice thickness at arbitrary coordinates, enabling the creation of distributed ice thickness maps for glaciers worldwide. The model operates using a set of 39 numerical features; its parameters are optimized globally. The model error is similar to state-of-the-art models in mid-to-low latitude glaciers, and up to 30-40% lower at high latitudes. However limited, the comparison with BedMachine demonstrates the skills of the machine-learned approach also in the ice sheet peripheries. As typical of machine learning methods, the model performance is expected to improve by increasing the training dataset size. Data from future measurement campaigns should be integrated into the training dataset. The large amount of training data available at high latitudes and the model errors in these regions suggest that, for our modeling approach, providing more data is more beneficial than providing more accurate data. Providing supervision (i.e. measurements) further reduces the model error by roughly a factor $\simeq 2$ to 3. Measurement campaigns targeting deep ice zones would prove extremely beneficial for improving IceBoost estimates of ice volumes. However, we find that not all glaciers benefit equally from added supervision on an individual basis. With the exception of DEMs which are available at high resolution and increasing accuracy, our modeling approach is not hard-constrained by the availability of specific input feature, notably ice velocity. Ice velocity improves the model by up to 6% at high latitudes, though no improvement is found elsewhere. Despite its marginal impact, this area holds the majority of the Earth ice volume. The most informative features are the distance to ice-free regions, surface slopes, surface curvature, and metrics of glacier size. An improved mass

balance feature will likely improve the model performance. We consider that our current local mass balance feature is only a simplified estimate. Our machine-learning approach is fully data-driven, with its primary advantage being the ability to learn directly from data. However, deeper insights can be achieved by integrating physical principles into machine-learning systems. Research in this direction would be a logical step forward.

*Code and data availability.* IceBoost source code is released on GitHub: https://github.com/nmaffe/iceboost. IceBoost trained modules (XG-Boost and CatBoost) are deposited on Zenodo as .json and .cbm files, respectively: https://doi.org/10.5281/zenodo.13145836. On Zenodo we also archive the Supplementary material: the training dataset, the model outputs for selected glaciers, alongside the comparisons with other models discussed in the text.

*Author contributions.* N.M. conceived and designed the research. N.M., E.R., S.V., T.P contributed to the development of the input feature set. N.M. wrote the model pipeline, performed the experiments and analyzed the results, with inputs from all authors. N.M. drafted the manuscript, to which all the authors contributed.

*Competing interests.* The authors declare that they have no conflict of interest.

*Acknowledgements.* We would like to thank Gianluca Lagnese and Fabien Maussion for valuable and constructive discussions and comments. We thank the following students from the Niels Bohr Institute - Applied Machine Learning 2024 class for their valuable ideas, analyses and parameter optimization: Emma Hvid Møller, Marcus Benjamin Newmann, Cerina von Bruhn, Jonas Richard Damsgaard, Josephine Gondán Kande, Luisa Elisabeth Hirche and Simon Wentzel Lind. This work has received funding from the European Union's Horizon 2020 research and innovation programme, under the Marie Skłodowska-Curie grant agreement No 101066651. This work was also funded by the Climate Change AI Innovation Grants program, hosted by Climate Change AI with the additional support of Canada Hub of Future Earth, under the project name ICENET.

## Appendix A: Training features

### A1 $Lmax$, distance from ice free regions ($d_{noice}$)

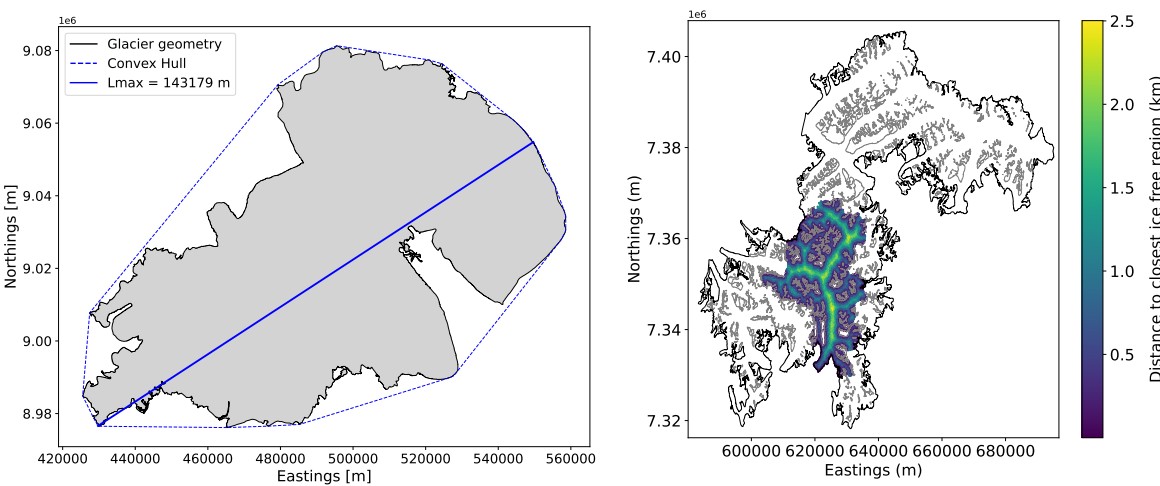

**Figure A1.** Left: 'RGI60-10315' glacier length ($L_{max}$). Right: 'RGI60-05.13995' glacier feature $d_{noice}$. The cluster external geometry with ice divides removed is show in black. All cluster nunataks are shown in grey.

### A2 Mass balance

#### A2.1 Polar ice sheet peripheries

In addition to glacier-averaged mass balance data from Hugonnet et al. (2021), we inform the model with local mass balances values. For the Greenland and Antarctic peripheries, we leverage the RACMO2 (Noël et al., 2018) product versions, downscaled, respectively, to 1 km (Noël and van Kampenhout, 2019) and 2 km (Noël et al., 2023). Before linearly interpolating the mass balance fields, we i) compute the time average over the 1961-1990 and 1979-2021 time periods respectively, and ii) fill some gaps in the dataset by convolving with Gaussian kernels of 1 km and 2 km respectively. Few gaps still remain in the mass balance fields, along some areas and glaciers (sub Antarctic islands and a few glaciers off the coasts of the Antarctic peninsula) not covered by these datasets. For these areas, as well as for all other glaciers, we use the approach described below.

#### A2.2 Glaciers outside polar ice sheets

For all glaciers outside the Greenland and Antarctic peripheries, we use the 2000-2020 mean glacier-integrated mass balance values from Hugonnet et al. (2021) and estimate the local variability by downscaling using approximate elevation-mass balance

rates. In particular, for all glaciers within the same region, we assume a linear variation of mass balance with elevation:

$$y = s \cdot h + q \tag{A1}$$

where $y$ is the mass balance and $h$ is the elevation. $s$ expresses the rate of change of mass balance with elevation, while $q$ reflects the mass balance at zero elevation. For any pairs of glaciers:

$$y_1 = s_1 \cdot h_1 + q_1 \tag{A2}$$

$$y_2 = s_2 \cdot h_2 + q_2 \tag{A3}$$

By using the glacier mean values $mb = \bar{y}$ from (Hugonnet et al., 2021) and further assuming that for close glaciers $s_1 = s_2 = m$ and $q_1 = q_2 = q$, we obtain:

$$s = \frac{mb_1 - mb_2}{\bar{h}_1 - \bar{h}_2} \tag{A4}$$

$$q = mb_1 - s\bar{h}_1 = mb_2 - s\bar{h}_2 \tag{A5}$$

For a given a glacier $i$, compute its mean rate $s_i$ by extending the calculation in Eq. A4 to all the other glaciers in the region $j \neq i$, weighting the mean by the inverse of the glacier distances:

$$s_i = \frac{\sum\limits_{i \neq j} \frac{\Delta mb_{ij}}{\Delta h_{ij}} \cdot \frac{1}{d_{ij}^2}}{\sum\limits_{i \neq j} \frac{1}{d_{ij}^2}} \tag{A6}$$

$$q_i = mb_i - s_i \bar{h}_i \tag{A7}$$

where $\Delta mb_{ij} = mb_i - mb_j$ and $\Delta h_{ij} = \bar{h}_1 - \bar{h}_2$ are the differences in glacier mass balance and average elevation between glacier $i$ and some glacier $j$, while $d_{ij}$ is the distance between the two glacier center values.

As an example, the distribution of $(s_i, q_i)$ calculated for all glaciers in RGI 11 (Central Europe, 3927 glaciers) is shown in Figure A2.

To compute mass balance maps for each glacier in each region, we use the regional mean values $\bar{s}$ and $\bar{q}$, listed in Table A1.

| RGI | 1 | 2 | 3 | 4 | 5 | 6 | 7 | 8 | 9 | 10 |
|---|---|---|---|---|---|---|---|---|---|---|
| $\bar{s}$ ($mm\ w.e.\ yr^{-1}m^{-1}$) | 0.46 | 0.34 | 0.22 | 0.65 | 0.55 | 0.84 | 0.86 | 0.56 | 0.64 | 0.30 |
| $\bar{q}$ ($mm\ w.e.\ yr^{-1}$) | -1038 | -1019 | -485 | -879 | -703 | -1082 | -524 | -1088 | -405 | -1034 |
| RGI | 11 | 12 | 13 | 14 | 15 | 16 | 17 | 18 | 19 | |
| $\bar{s}$ ($mm\ w.e.\ yr^{-1}m^{-1}$) | 0.41 | 0.14 | 0.46 | 0.16 | 0.32 | 0.52 | 0.45 | 0.14 | 0.41 | |
| $\bar{q}$ ($mm\ w.e.\ yr^{-1}$) | -1739 | -919 | -1956 | -919 | -2054 | -2889 | -1051 | -440 | -191 | |

**Table A1.** Regional values of $\bar{s}$ and $\bar{q}$.

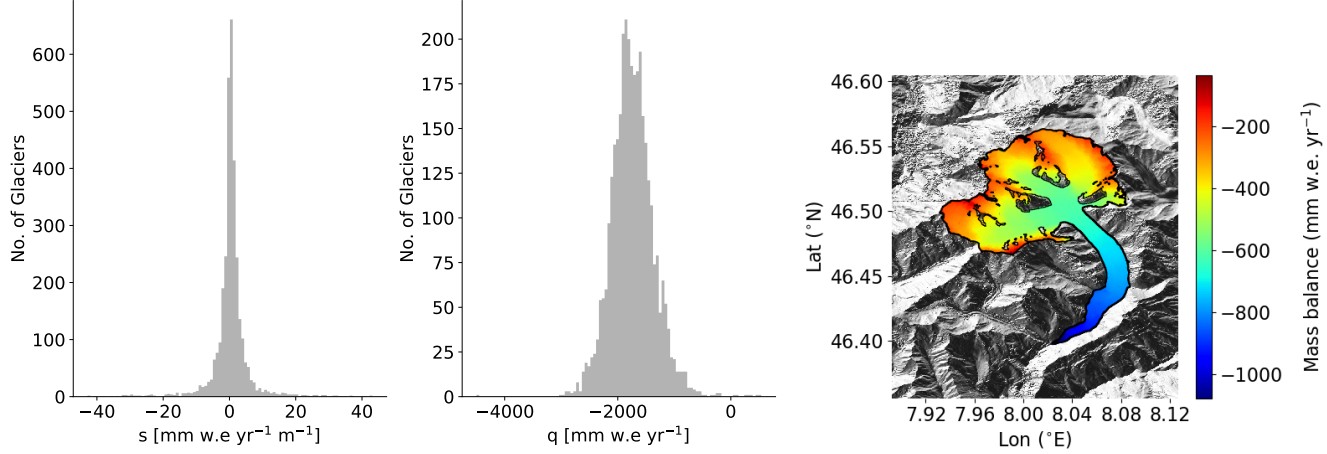

**Figure A2.** Distribution of $(s_i, q_i)$ for RGI 11 along with the mass balance distribution for the Aletsch glacier.

Using this method we can replicate the glacier-integrated mass balance values (Hugonnet et al., 2021) within a factor $\approx$ 2-3. Given all the hypotheses made, we consider our downscaling approach as an attempt to provide the model with crude, yet local, mass balance approximations. An analysis of the impact of uncertainties in $(\bar{s}, \bar{q})$ on the modeled glacier volumes is presented in Appendix D.

### A3   IceBoost hyperparameters

The best XGBoost hyperparameters found during the Bayesian optimization pipeline are: tree_method=hist, lambda=76.814, alpha=76.374, colsample_bytree=0.9388, subsample=0.741501, learning_rate=0.079244, max_depth=20, min_child_weight=19, gamma=0.18611. We use 1000 trees (num_boost_round) with early_stopping_rounds=50. For CatBoost: iterations=10,000, early_stopping_rounds=50, depth=6, learning_rate=0.1. For the parameter description, we refer to the XGBoost documentation at https://xgboost.readthedocs.io/en/stable/parameter.html and CatBoost at https://catboost.ai/en/docs/concepts/parameter-tuning.

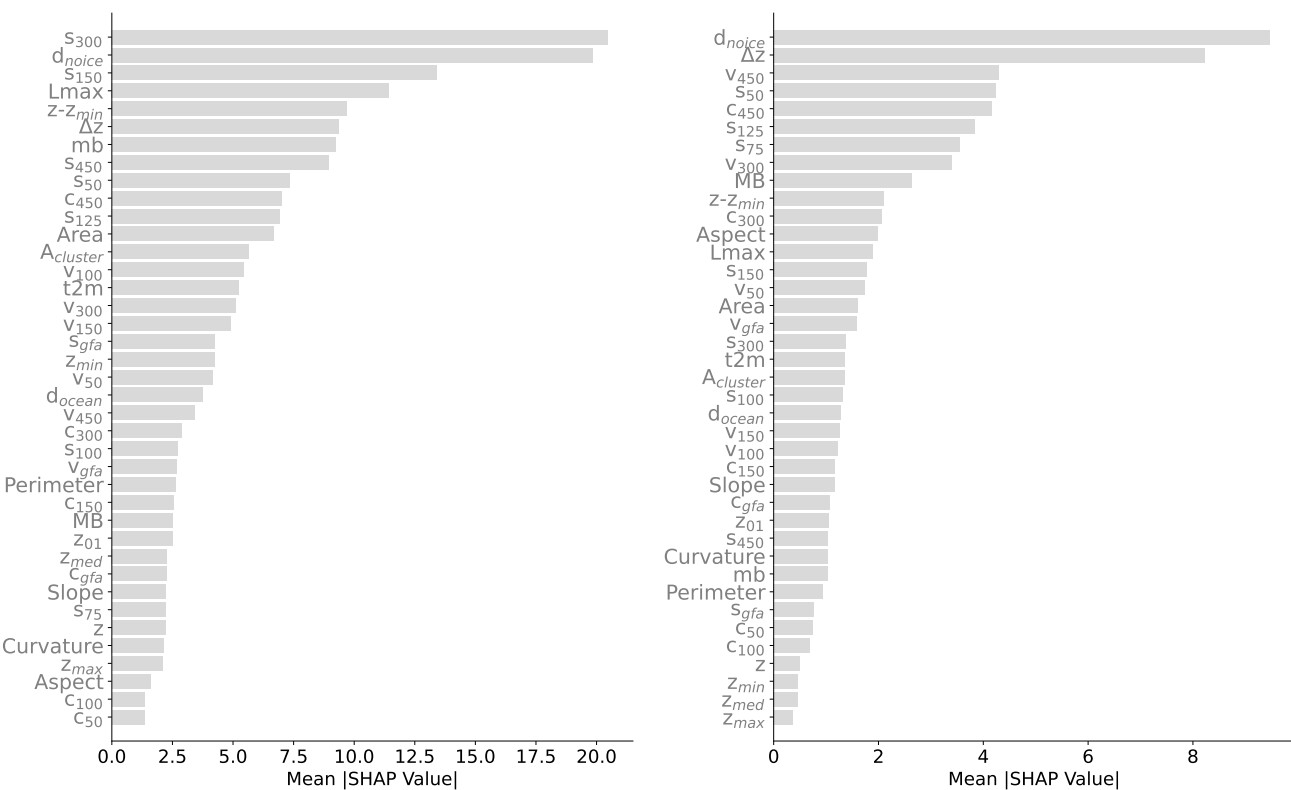

**Figure A3.** Feature ranking: the ranking of each feature is taken to be the mean absolute shape value for that feature over n=2,000 random samples (left) and over n=2,000 samples from Central Europe (RGI 11, right).

## Appendix B: Model inference

### B1 Fetching features on-the-fly

At inference time, the features are generated on-the-fly following the same pipeline described for the creation of the training set. As an example, Figure B1 shows the extraction of the $v_{50}$ feature for n=1500 random points.

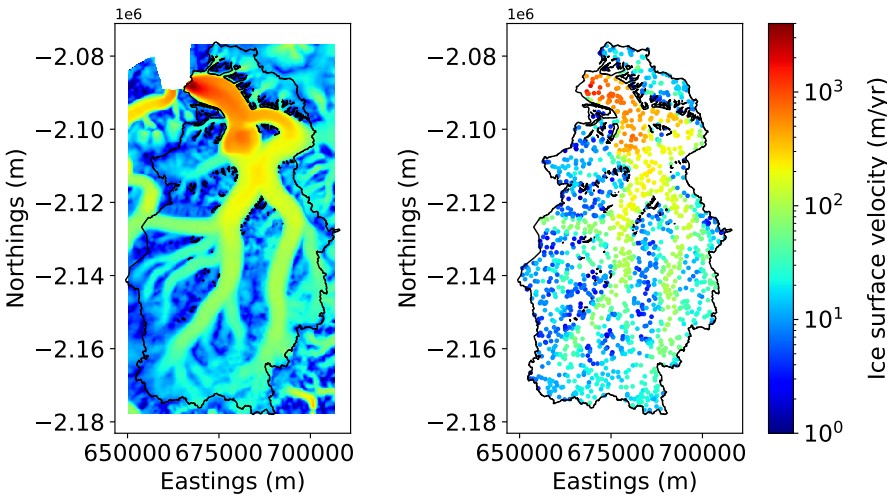

**Figure B1.** Pipeline for feature generation at inference time. Left: ice velocity ($v_{50}$, from Joughin et al. (2016) over glacier RGI60-05.13501 in East Greenland. Right: feature calculated for n=1500 random points.

### B2 Feature imputation policies

Feature imputation is need whenever any feature is not available either for the creation of the training dataset or at inference time. Unless specified, we adopt the same policy in both cases. Hereafter is a list of the features that may require imputation and their imputation policies.

Ice velocity

No imputation is implemented at training time: if any velocity feature is missing at any point, the point is not included in the training dataset. This condition occurs if the training point falls outside the velocity field (old measurement or measurement inside a nunatak or incomplete velocity coverage) or if it is too close to the geometry such that the interpolation fails. At inference time, a complete velocity feature coverage is required as input for the model. A 3-layer progressive policy is implemented to fill any missing data and ensure complete coverage of all velocity features: i) kernel-based interpolation using a Fast Fourier Transform convolution and Gaussian kernels, ii) glacier-median imputation and iii) regional-median imputation.

Mass balance

- Glacier-mean values: for glacier ids listed in RGI v. 62 but not present in Hugonnet et al. (2021) mass balance dataset, which is tied to RGI v.6 glacier dataset, we impute a regional median value. RGI v7 glaciers, the product by Hugonnet et al. (2021) is not available. To be able to use it, we link all glacier ids from RGI v6 to RGI v7, by finding the glacier in RGI v7 that has the maximum area overlap with any RGI v6 glacier. If no glacier is found, we impute a regional median value from RGI v6.

- The RACMO products used for Greenland and Antarctica do not cover some glaciers located on islands proximal the ice sheets. These include almost all glaciers from the sub-Antarctic islands. For these, we use the downscaling approach described in Appendix A2.2.

## B3 Glacier volume calculation

The glacier volume is approximated by Monte Carlo as $V_{gl} = A_{gl} N^{-1} \cdot \sum_{x,y} h(x,y)$, where $A_{gl}$ is the glacier area, $h(x,y)$ is the modelled thickness at point $(x,y)$ inside the glacier, N is the total number of generated points. This method, tested by comparing Farinotti's interpolated thickness values against their true values allows to estimate the Monte Carlo error to less than 1%, even for the biggest glaciers. While $N = 10^4$ allows for a precise volume estimate, to better evaluate the spatial variability of the solution over scales of tens of meters, N can be increased to $O(10^5)$, depending on glacier size, or increased locally to target specific regions.

## B4    Comparison with BedMachine Greenland

Figure B2 shows a comparison between IceBoost and BedMachine v5 (Morlighem, 2022b) for a glacier with direct connection
to the ice sheet. Note the additional complexity of the fjord system predicted by IceBoost, compared to BedMachine. While
an extensive comparison with BedMachine is beyond the scope of this work, we highlight the potential of IceBoost in the ice
sheet peripheries.

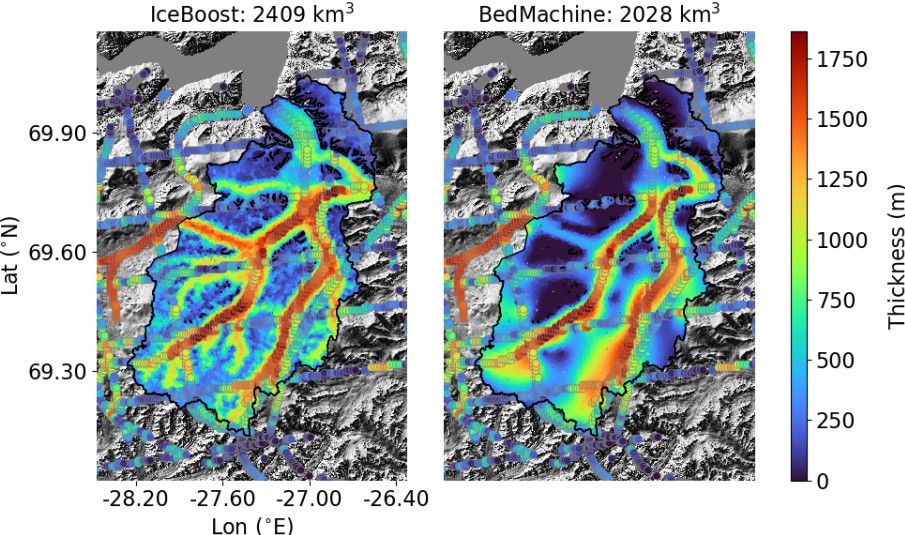

**Figure B2.** Glacier RGI60-05.13501 modeled by IceBoost, and BedMachine v5 (Model1, Morlighem 2022b).

## B5    Artifacts in the modeled output

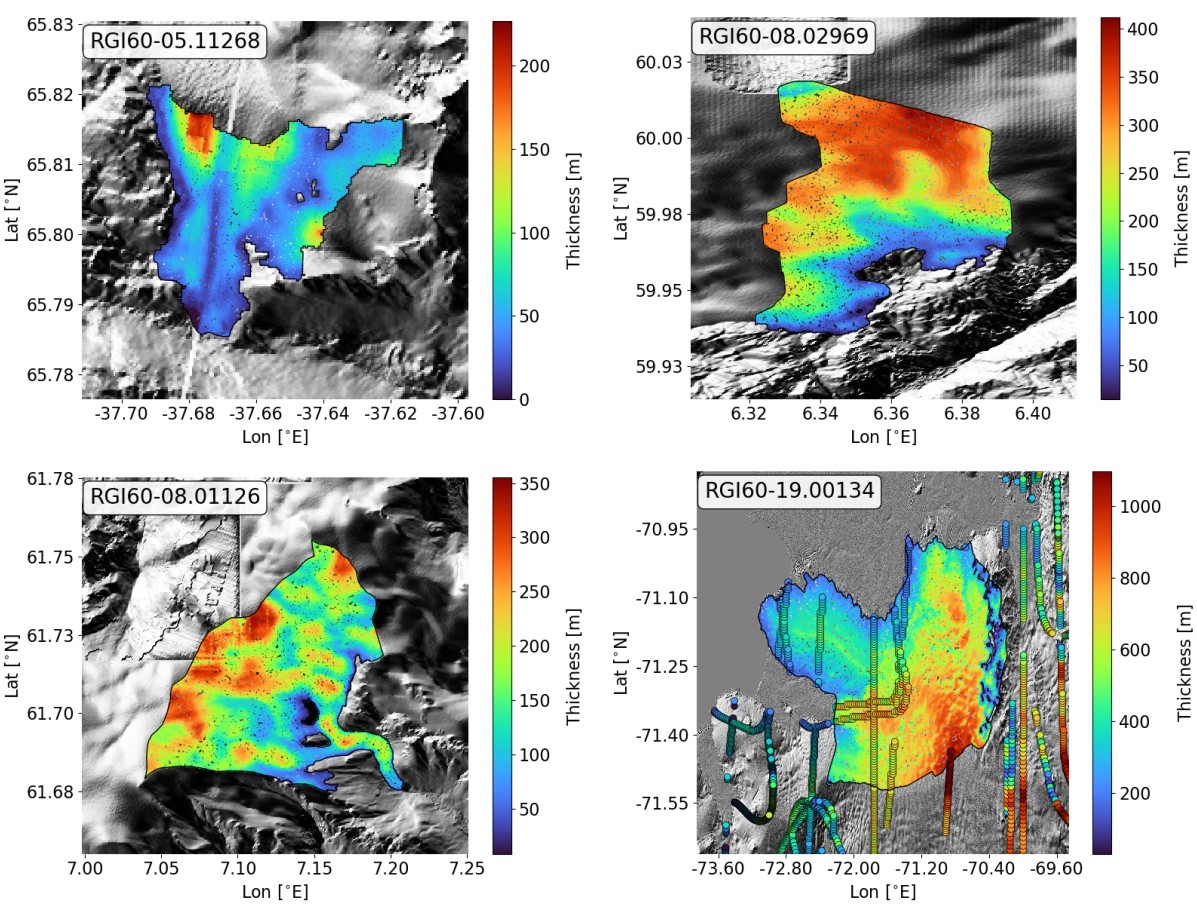

**Figure B3.** Modeled glacier ice thickness. Artifacts in the Tandem-X DEM are visible for glaciers RGI60-05.11268, RGI60-08.02969, and RGI60-08.01126. The effects of such artifacts is visible in the modeled thickness map. Glacier RGI60-19.00134 modeled thickness shows some high frequency noise, which mimics the roughness of the surface.

## Appendix C: Sensitivity of model outputs to input glacier-integrated mass balance

We assess the model's sensitivity to input mass balance values (feature *MB*, imported from Hugonnet et al. 2021) using the Unteraargletscher (Switzerland, 46°34'0"N 8°13'0"E) as a case study. This glacier was chosen due to extensive surveys conducted by Swiss glaciologists over decades, and ground truth measurements. The reference mass balance value is -1.59 m w.e yr$^{-1}$ (Hugonnet et al., 2021). IceBoost is run over a range of mass balance values from -4.0 to 4.0 m w.e yr$^{-1}$, extending beyond realistic limits for this glacier, and including the reference value -1.59 m w.e yr$^{-1}$ (Fig. C1). Glacier volume shows limited

dependence on mass balance, varying by at most 20% across the entire range, with some predictions differing by less than 0.001 km$^3$. In our model setup, glacier-integrated mass balance is a weak predictor of local ice thickness. This finding aligns with global results obtained from the variable ranking analysis on the training dataset (Sect. A4, Fig. A3).

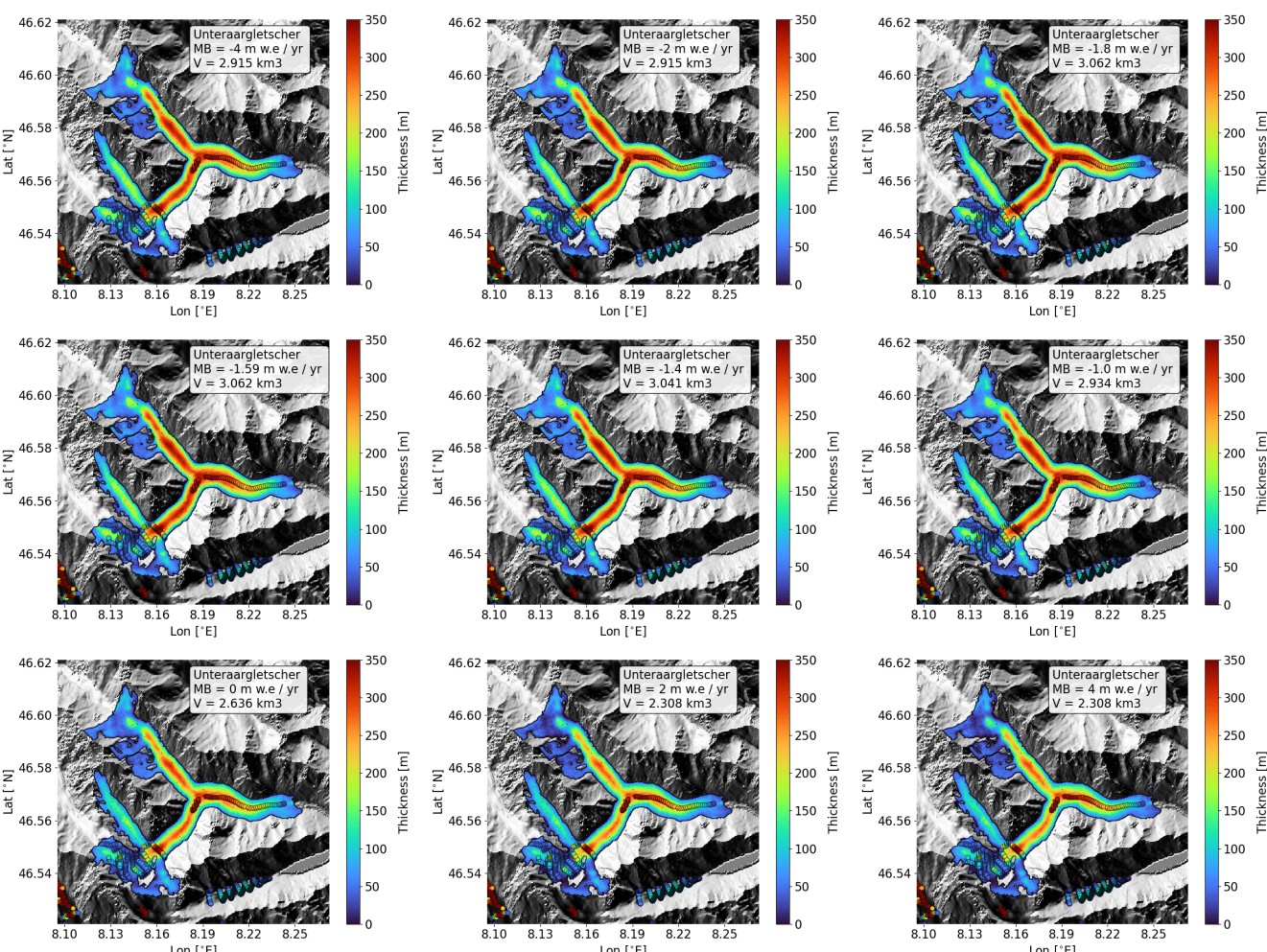

**Figure C1.** Unteraargletscher ice thickness modeled with different input mass balance values, indicated in the insets. The reference mass balance for this glacier is -1.59 $m\ w.e\ yr^{-1}$ (Hugonnet et al., 2021). Circles refer to ground truth data.

## Appendix D: Sensitivity of model outputs to input spatially distributed mass balance

The uncertainty in the inferred regional values of $(\bar{s}, \bar{q})$ used to calculate mass balance maps (feature mb, see Appendix A2.2) is investigated here. These parameters represent, respectively, the rate of change of mass balance with elevation ($\bar{s}$), and the mass balance at zero elevation ($\bar{q}$). The parameters are calculated regionally, hence all glaciers in the same region share the same values.

We analyze two glaciers in Switzerland: Unteraargletscher (46°34'0"N 8°13'0"E) and Aletsch (46°26'32"N 8°4'38"E). For both glaciers, $\bar{s} = 0.41$ mm w.e yr$^{-1}$m$^{-1}$, $\bar{q} = $ -1739 mm w.e yr$^{-1}$ (Appendix, A2.2). We conduct three Monte Carlo simulations with n=500 iterations. For each set of simulations, random noise drawn from a Gaussian distribution is added to the $\bar{s}$ and $\bar{q}$ input variables, with widths of respectively 10%, 20% and 50% their nominal values. The resulting distributions of modeled output volumes are shown in Figure D1, compared to the nominal values of 3.06 km$^3$ (Unteraargletscher) and 14.4 (Aletsch) km$^3$, obtained without added noise.

For a 50% uncertainty in $\bar{s}$ and $\bar{q}$, the modeled glacier volume changes, on average, by 9.5% (Unteraargletscher) and by 2.2% (Aletsch), with a variability of 14% and 9.5%, respectively. The total uncertainty, combining systematic and random components, is ±17% for Unteraargletscher and ±9.8% for Aletsch. With a 20% uncertainty, the volume error is ±8.2% (Unteraargletscher) and ±6.4% (Aletsch), while for a 10% uncertainty it is ±3.7% and ±0.5%, respectively.

This sensitivity test, though limited to two glaciers in Central Europe, suggests that the error in the modeled glacier volumes due to uncertainty in mass balance parametrization can be safely considered to not exceed 15%.

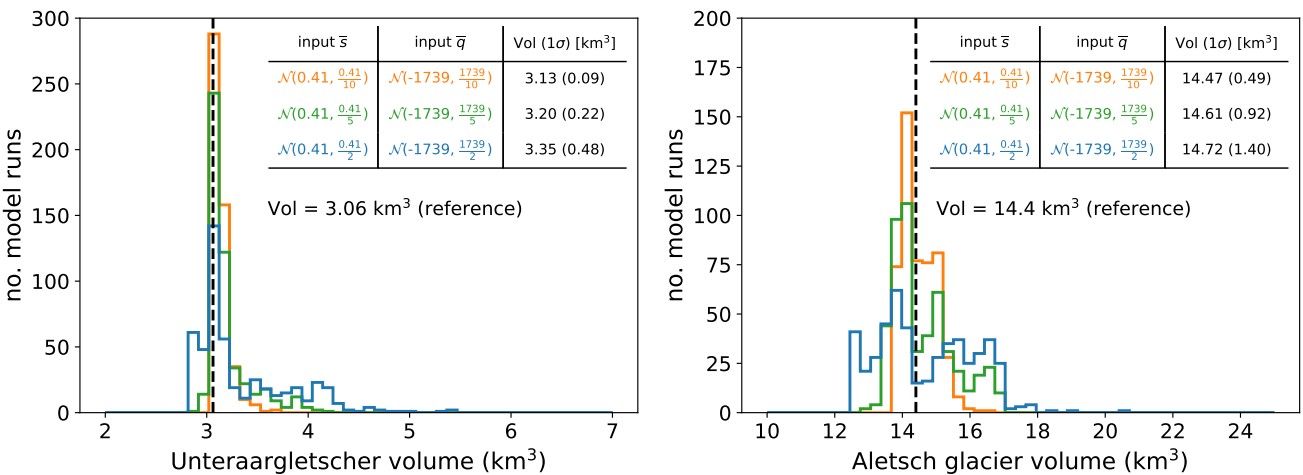

**Figure D1.** Monte Carlo simulations of modeled glacier volumes for Unteraargletscher and Aletsch, incorporating Gaussian noise in the input parameters ($\bar{s}$, $\bar{q}$). Noise levels of 10% (orange), 20% (green), and 50% (blue) are applied. These parameters are used to compute spatially distributed mass balance (Appendix A2.2). $\bar{s}$ is expressed mm w.e yr$^{-1}$m$^{-1}$ and $\bar{q}$ in mm w.e yr$^{-1}$. The black line represents the modeled glacier volume without added noise.

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
