# Peer review of "A gradient-boosted tree framework to model the ice thickness of the World's glaciers (IceBoost v1.1)"

_EGUsphere, 2024_

## Referee Comment (RC1)

**Review of 'A gradient-boosted tree framework to model the ice thickness of the world's glaciers (IceBoost v1)' by Maffezzoli et al.**

**December 2, 2024**

In 'a gradient-boosted tree framework to model the ice thickness of the world's glaciers (IceBoost v1),' the authors present a method based on random forests for estimating ice thickness globally. The method differs substantially from more classical methods for ice thickness estimation in that it does not utilize principles of glacier-wide mass conservation. Instead it is trained on a large number of geometric and other variables calculated locally. The results are encouraging, especially when the model is trained on data from the same glacier on which it is making predictions.

I think this is an interesting paper that may be useful to some practitioners who are interested in estimating global glacier volume. The methods make sense and it is clear what is being done. I would like to see a somewhat more comprehensive description of some of the method's shortcomings. For example, I think a better job could done of trying to understand in what circumstances the method performs well versus not – the Malaspina example could do this, but instead we just get 'works okay here, but look at this other example where it works better!'. I also would like to see at least a discussion of the suitability of the resulting product for different tasks, e.g. as input for modelling, which this will invariably be used for, whether its suitable or not. It would be nice to hear a bit about predictive uncertainty - even if it's just to say that you don't know because the model doesn't produce that. As a small concern, it would be helpful to add a section describing why – not just what – features were included in the analysis. It's not immediately clear why having slope smoothed over 8 different length scales (some of which aren't so different) is a good idea. Finally, after all the fuss made over this being globally applied, why not provide an estimate of global ice volume and compare it to previous works?

Detailed comments are below.

- L3 Here and elsewhere, I think the word 'distribution' needs to be used with more precision. In particular, I think it would be better to say that the method evaluates the thickness at arbitrary coordinates and can produce a map.
- L28 A comment on this section it's not necessarily the case that other methods for the 'global' problem don't exist, it's just that many people working in this field don't believe that it's defensible to produce such estimates because input observations e.g. surface mass balance are not reliable.
- L24–39 It is worth noting potential limitations of these methods (of which there are many). What does the present work bring to advance beyond these previous efforts?
- L46–48 This is an awkward sentence, and it won't be clear to the lay reader what the distinction is between image and tabular data, particularly since we almost always look at thickness as maps (i.e. images).
- L57 The absolute number of training data 'points' isn't all that relevant because they are adjacent to one another and many (perhaps even most) of these data points are strongly correlated with one another.
- L60 'The DEM choice ...' I don't really know what this line means.

- L61 I suggest being more explicit about what these features are. For example, instead of calling RACMO 'mass balance information', it would be more helpful to describe it as 'modelled estimates of spatially-distributed surface mass balance'.
- L77 I think that this line about BedMachine will only make sense to someone who is already very familiar with how BedMachine is constructed.
- L110 Symbols are arbitrary of course, but glaciological literature usually uses H (or sometimes (h) to represent ice thickness.
- L113 This decision makes sense because you include features that already encode this regional variability (SMB, elevation, etc.). It would be weird to include these things and then also need regionally stratified models.
- L121-L136 I don't think this section does a good job of justifying not holding out a validation set after doing 200 trials, the hyperparameter choices ought to be robust to randomness in the splits because there are so many of them. If these hyperparameters don't perform well on another independent data set, then the model isn't very good.
- L137 I don't think 'accuracy' and 'precision' should be used here. Just state the metrics that are used (median residual and RMSE). Also, why these two?
- L146 I think I understand what this paragraph is saying, but I would like to see this clarified. Are regions exlcuded from analysis because there's no data or because the performance isn't good? What does it mean for performance to be 'indicative'?
- L150 This is an awkward jump to start talking about the combined models when there has yet to be any description of the performance of the individual models.
- L153 The more cynical take regarding the better performance of the 'supervised' model is that the model is overfit and only performs well when it can look up memorized observation points that are close in space to the query.
- L161 Shouldn't this appear in some kind of 'data availability' statement?
- L209 Malaspina is currently land terminating, so its terminus is mostly grounded at or above sea level. When using 'terminus' do you mean the piedmont lobe?
- L227 I am not sure how this argument is justified. Is it possible to be more quantitative?
- L234 'Conversely, the features not based on satellite products are not discrete'. I don't understand this sentence.
- L236 There are fundamental limits to the resolution of bed features that can be determined solely from surface observations due to the diffusive nature of ice flow.
- L267 These conclusions are fine, but please write them in narrative form, rather than as bullet points.

---

## Author Response (AR1)

**General Model and Data updates**

Since the release of model version v.1 at the beginning of the discussion (September 23, 2024), we have made the following updates:

**Ground Truth Data used for model training**

1. **Zero-Thickness Measurements in GlaThiDa**
   A substantial number of zero-thickness measurements were identified in GlaThiDa across regions such as the Canadian Arctic, Greenland (the largest proportion), Svalbard, and Antarctica. While some of these measurements occur near glacier boundaries, others are located within glaciers (see also GitLab issue #25), where ice thickness should not be zero. These measurements have now been removed.

2. **Additional Measurements from IceBridge**. Approximately 11,000 measurements from 44 glaciers, not included in GlaThiDa, have been added from the IceBridge MCoRDS L2 Ice Thickness, Version 1.

3. **Error in GlaThiDa glacier 'RGI60-19.01406'**. Ground truth data for glacier *RGI60-19.01406* (a peripheral glacier in Antarctica, located at 65.5°S, 100.8°E, maximum elevation 500 m a.s.l.) is listed with with ice thickness values likely with a factor ten too much (ice exceeding 3,000 meters). We divide this data by 10.

**Feature set**

All features previously imported from RGI are now directly calculated from the DEM or from the glacier geometry: *glacier area*, *glacier length*, *minimum, maximum, and median glacier elevation*, as well as *glacier average slope*. The features *TermType*, *Form*, and *local aspect* have been dropped. We compared the newly calculated features with their counterparts in RGI6 and found minimal or no differences. During inference, all features continue to be calculated on-the-fly, as before. These modifications offer the following benefits:

- The feature set and model can now adapt to any DEM, eliminating reliance on the Randolph Glacier Inventory (RGI). However, the model still depends on RGI for glacier geometries and their connectivity.

- The feature set is now self-contained, removing the need for imputation policies for features previously extracted from RGI (e.g., *Lmax* and *zmed*, which had significant missing data).

The *TermType*, *Form* and *local Aspect* features have been dropped, as previous analyses showed they provide minimal informational value.

**We added the following features:**

- **Cluster Area**, $[km^2]$: The sum of all glacier areas within a maximum graph connectivity depth of three. For a given glacier $\hat{x}$, this includes all glaciers in the graph $G(vertices = \{y_i, z_i, w_i\}, edges)$ connected with edges $E = \hat{x} \rightarrow y_i \rightarrow z_i \rightarrow w_i$. The calculation utilizes the Python NetworkX library. This feature indicates whether a glacier is isolated ($A_{glacier} = A_{cluster}$) or part of a larger system. It is (weakly) positively correlated with ice thickness, as found from the Shapely analysis.

- **Glacier Mean Curvature**, $[0.01 \ m^{-1}]$. The mean curvature is calculated over the entire glacier area.

- **Glacier Mean Slope**. The mean slope is calculated over the entire glacier area.

- **Normalized local elevation**, ($z_{01}$): $z_{01} = \frac{z - z_{min}}{z_{max} - z_{min}}$

- **Local Curvature variables**: Represented by six variables corresponding to six kernel sizes: 50 m, 100 m, 150 m, 300 m, 450 m, and an adaptive filter based on glacier area (similar to elevation and ice velocity). The number of curvature variables was increased due to their relatively high importance, as shown by SHAP analysis.

We note that several features, particularly those calculated using the Python Xarray-Spatial library, can be accelerated on GPU via RAPIDS. While support for GPU acceleration will be implemented in a future model version, a GPU is not currently deemed necessary. This is because, for small glaciers (the majority), the time required to transfer data to GPU memory may exceed the time needed for feature calculation on the CPU.

Significant improvements have been made to the code efficiency. With the latest modifications, a glacier solution is now produced in 1–2 seconds on average. In a multi-CPU setup ($n_{jobs} = 8$), all glaciers in Svalbard (n=1,615) can be processed in approximately 12 minutes.

**Other changes**

- All figures have been updated by re-running the revised model. SHAP plots now reflect the updated feature set, and Figures 3, 4, and 7 have been revised accordingly.

- Table 2 values have been adjusted to account for changes in the feature set. The conclusions remain unchanged.

- We add a Shapely analysis of the Mittie Glacier (as a case study), demonstrating the utility of SHAP values in understanding feature importance for individual glaciers. We believe this is an important aspect, as allows to understand the amount of information and the value of all features on a glacier-basis.

- All thickness plots now use the colorblind-friendly 'turbo' colormap, replacing 'jet'.

- Support for the latest RGI v.7 glacier geometries has been added. The model continues to use the OGGM python library as a convenient tool for importing RGI geometries.

- All comparisons with the other two models (n=190 glaciers) (previously deposited on Zenodo), are now also included in a Supplementary Information PDF for convenience. The file size is approximately 30 MB.

**Referee no. 1, December 2, 2024**

In 'a gradient-boosted tree framework to model the ice thickness of the world's glaciers (IceBoost v1),' the authors present a method based on random forests for estimating ice thickness globally. The method differs substantially from more classical methods for ice thickness estimation in that it does not utilize principles of glacier-wide mass conservation. Instead it is trained on a large number of geometric and other variables calculated locally. The results are encouraging, especially when the model is trained on data from the same glacier on which it is making predictions.

We thanks Referee no. 1 for taking the time to review our manuscript. Please find our answers in blue. For clarity, we point out that the model is an ensemble of gradient-boosted decision trees, not based on random forests.

I think this is an interesting paper that may be useful to some practitioners who are interested in estimating global glacier volume. The methods make sense and it is clear what is being done. I would like to see a somewhat more comprehensive description of some of the method's **shortcomings**. For example, I think a better job could done of trying to understand in what circumstances the method performs well versus not – the Malaspina example could do this, but instead we just get 'works okay here, but look at this other example where it works better!'. I also would like to see at least a discussion of the suitability of the resulting product for **different tasks**, e.g. as input for modelling, which this will invariably be used for, whether its suitable or not. Thanks. We now added a section "Applications, improvements, and limitations" [main text, Sect. 5, page 19], where we discuss the shortcomings and some possible applications. We have added some figures in the Appendix [Appendix B3, page 29] that show produced artifacts.

It would be nice to hear a bit about **predictive uncertainty** - even if it's just to say that you don't know because the model doesn't produce that. The IceBoost model does not come with any uncertainty estimate, though this can in principle be produced by other algorithms (e.g. NGBoost). This has been now been added in the text, after the discussion of the feature SHAP values [page 17, line 358], as well as in the improvements section.

As a small concern, it would be helpful to add a section describing why – not just what – features were included in the analysis. It's not immediately clear why having slope smoothed over 8 different length scales (some of which aren't so different) is a good idea. Please also note this was also a concern of Reviewer no. 2. We now expanded the description of the features, by dedicating a separate section (Sect. 2.2, pages 4-8). We motivate their choice, and added references. On the specific point of slope smoothing, different methods can be adopted. The best kernel size likely depend on individual glaciers in relation to its mean ice thickness. For example, Millan et al. (2022) adopt a kernel of variable size depending on the distance from the glacier border. Zorzut et al. 2020 investigated the effect of kernel size on one specific glacier, and found an optimal value of 300 meters in their experiments. Historically, kernels of up to 10-20 times high thickness have been used. As we develop one global model with glacier size spanning many orders of magnitude, we instead use many kernels let the model decide which ones to use for every individual glacier. From the SHAP analysis we found that indeed the model uses different kerels in different scenarios. For example, for small glaciers in the European Alps (RGI 11, Fig. A3), smaller kernels are found to have higher ranking. The opposite occurs for large ice caps, where the model prioritizes larger kernels. Other choices could be certainly done, for example, using one single kernel of variable size (the difficulty being the choice of the functional form to size them). See also our reply to Referee 2, point 1. The kernel sizing is now mentioned in the section "Applications, improvements, and limitations", as it is indeed challenging.

Finally, after all the fuss made over this being globally applied, why not provide an estimate of global ice volume and compare it to previous works? We find this appropriate. However, it

is the scope of this manuscript to present the model and its global range of use. Global deployment takes considerably more effort with further analyses and careful assessments certainly needed, and will therefore be discussed at a later stage.

Detailed comments are below.

**L1** Here and elsewhere, I think the word 'distribution' needs to be used with more precision. In particular, I think it would be better to say that the method evaluates the thickness at arbitrary coordinates and can produce a map. We agree. Here and in the conclusions, we clarify and rephrase as "IceBoost, a global machine learning framework trained to predict ice thickness at arbitrary coordinates, thereby enabling the generation of spatially distributed thickness maps for individual glaciers".

**L28** A comment on this section - it's not necessarily the case that other methods for the 'global' problem don't exist, it's just that many people working in this field don't believe that it's defensible to produce such estimates because input observations - e.g. surface mass balance - are not reliable. The comment is well received. We rephrased to "Only two have been applied to all glaciers on Earth."

**L24–39** It is worth noting potential limitations of these methods (of which there are many). What does the present work bring to advance beyond these previous efforts? We now expanded on the limitations of previous methods, by invoking the challenges with respect to i) the physical assumptions, ii) the challenges in calibrating the model parameters, and iii) quality and availability of input data. We outline these consideration by taking the shallow ice approximation as a case study (for brevity), while referring the readers to Farinotti et al. (2017) for an comprehensive overview on all methods. We also expanded on the challenges of deep learning approaches in the paragraph that follows. And finally, in the last paragraph when the model is introduced, we added some considerations of the advantages/advancements that we think this work brings.

**L46–48** This is an awkward sentence, and it won't be clear to the lay reader what the distinction is between image and tabular data, particularly since we almost always look at thickness as maps (i.e. images). Thanks for the comment. There is a substantial difference in model choice when training a ML model using tabular vs gridded data. The ice thickness target data is tabular data. The models that can "naturally" ingest tabular data are tree-based models not convolutional neural networks, which typically ingest images. To train a CNN on ice thickness, Lorenzo Lopez Uroz et al. (2024) had to first create an ice thickness gridded product by interpolating thickness measurements, effectively injecting interpolated information when training. We improved the phrasing of the whole paragraph to better clarify that the model choice is driven by the data type: *"IceBoost employs an ensemble of two gradient-boosted decision tree models, XGBoost and CatBoost, which are trained using ice thickness as target variable. The target data is naturally tabular-structured and is extracted from the Global Ice Thickness Database. The model is informed using a set of 39 numerical features, extracted from an array of products and similarly organized in a tabular structure. While deep learning methods are best suited to operate on gridded products (images), when data is tabular-structured (as are the localized-in-space measurements of ice thickness), tree-based models often provide a much faster and powerful alternative (Grinsztajn et al., 2022).*

**L57** The absolute number of training data 'points' isn't all that relevant because they are adjacent to one another and many (perhaps even most) of these data points are strongly correlated with one another. Thanks, we agree that correlation between training entries needs attention. In our work we adopt a spatial gridding strategy, to reduce correlation

between points collected close to each other and reduce the skewness of input variables and the target. We have investigated the different gridding size by checking the model performance at validation time. We find the best results with 100x100 bins. Eventually, a too coarse size (for example 20x20) results in too much information being discarded. In the data pre-processing section (L 97) we added that "To reduce the amount of correlated data close to each other and reduce computational costs .... ".

**L60** 'The DEM choice ...' I don't really know what this line means. The sentence has been deleted.

**L61** I suggest being more explicit about what these features are. For example, instead of calling RACMO 'mass balance information', it would be more helpful to describe it as 'modelled estimates of spatially distributed surface mass balance'. Done. Please also note that now the features are described in the stand alone section (Sect 2.2 Training features, page 4).

**L77** I think that this line about BedMachine will only make sense to someone who is already very familiar with how BedMachine is constructed. We do not mention BedMachine here anymore. When introducing BedMachine down below, we now include a brief explanation.

**L110** Symbols are arbitrary of course, but glaciological literature usually uses H (or sometimes (h) to represent ice thickness. y now changed to h. All elevations changed to $z$.

**L113** This decision makes sense because you include features that already encode this regional variability (SMB, elevation, etc.). It would be weird to include these things and then also need regionally stratified models. We agree. We points out that a global model can leverage the entire global dataset. The disadvantage is probably an increased complexity (in modeling different flow regimes, in using many input data products, ...). Even more explicit regional variables could theoretically be injected (e.g. latitude), but we decided not to, as they do not always carry physical meaning.

**L121-L136** I don't think this section does a good job of justifying not holding out a validation set - after doing 200 trials, the hyperparameter choices ought to be robust to randomness in the splits because there are so many of them. If these hyperparameters don't perform well on another independent data set, then the model isn't very good. Due to the heterogeneity of the global ice thickness ground truth data (a very non-uniform value distribution, Supp. Info. Fig 1) we found that evaluation of the model on the test set was strongly dependent on the data that felled into that split (typically small). As an example, if Antarctic data were part of the test set, the model error evaluated would be higher compared to the case in which Antarctic data would not be included, by mere stochasticity. For this reason, we did not reserve a test split. In addition, we aimed for a pipeline that used as much data as possible. In general, we explored different ways to tackle the heterogeneity of the data: a loss weighted for number of occurrences (penalizing for 1/frequency), leaving out over-represented glaciers or regions, or data augmenting the under-represented thickness values. All these methods proved not particularly successful.

The whole paragraph was modified for clarity. We also added a part in which we further motivated our choice of not producing a test set.

**L137** I don't think 'accuracy' and 'precision' should be used here. Just state the metrics that are used (median residual and RMSE). Also, why these two? We now mention only the metrics. The median of the residual distribution is used to assess for bias, while the RMSE is used to assess the model error. The latter is a common choice for regression

problems. We preferred using the RMSE over MAE (mean absolute error) because the model is trained with a squared loss, therefore a squared error is a very natural choice. The choice of using a quadratic loss for training was made to impose heavier penalties on large errors.

**L146** I think I understand what this paragraph is saying, but I would like to see this clarified. Are regions excluded from analysis because there's no data or because the performance isn't good? What does it mean for performance to be 'indicative'? We improved the phrasing. To clarify, we chose to report performance in Table 2 whenever enough data allowed us to do so with statistical significance (not based on performance results). For example, statistics for RGI 2 are not reported as only 2 glaciers are part of the training dataset (Fig. 1), and we cannot draw any regional conclusions on the model performance with only 2 glaciers. The same is valid for RGIs 6, 9, 14, 15, 16, and 18. Similarly, we acknowledged that few data exist in our training set for RGIs 10, 12, 13, and 17 and therefore we consider the statistics in those regions as indicative, and possibly similar to other regions where the ice flow regime is similar (e.g. the entire Himalaya regions, RGI 13-14-15). See SP1 from Reviewer2 for the updated phrasing.

**L150** This is an awkward jump to start talking about the combined models when there has yet to be any description of the performance of the individual models. We now include Table 1 in the Supplementary Information, which shows the performance of individual models across the different regions. We also show an example of individual outputs (Academy of Sciences ice cap), Supp. Info. Fig 2. We have now included an evaluation of the two individual models to begin with. Such paragraph reads: *The performance of XGBoost and CatBoost, evaluated individually on ground truth data, is comparable, with neither consistently outperforming the other within 1-sigma statistical fluctuations (Supp. Info., Table 1). A qualitative comparison at inference time on selected glaciers suggests the same conclusion holds, with similar predicted patterns even in the absence of ground truth data (Academy of Sciences ice cap shown in Supp. Info., Fig. 2). We create IceBoost by taking an unweighted mean of XGBoost and CatBoost. Alternative approaches, such as applying regional weighting or per-glacier weighting based on feature explainability, could be explored but are left for future work.*

**L153** The more cynical take regarding the better performance of the 'supervised' model is that the model is overfit and only performs well when it can look up memorized observation points that are close in space to the query. At validation stage we can only conclude that by adding supervision (on a regional basis) the error decreases significantly. Given the squared error, this improvement likely stems from better predictions of high-thickness data points (one reason of using a squared loss). Several regularization mechanisms have been enforced to prevent overfitting (see parameters lambda, alpha, colsample_bytree, subsample, gamma, and early_stopping_rounds in Appendix A3). When additional training data becomes available, the model naturally incorporates it, aligning with typical machine learning schemes and not necessarily indicating overfitting. This behavior parallels physics-informed neural networks, where ground truth data further constrains PDE solutions via an additional data loss term alongside the physics loss. In our case, the model may struggle with certain glaciers initially and benefit from ground truth data to enhance predictions. However, this is not always the case (luckily), as additional data is not always needed—unlike a systematically overfitting model. The same analysis at inference time on individual glaciers (not at validation stage, which is done regionally) offers additional insights. For example, in the Kongsbreen glacier modeled here below, probably not all ground truth data collected in the lower half of the glacier is necessary, if added.

Viceversa, and similar what we found for the Malaspina glacier, some discrepancies still remain in upper half, where the model still cannot replicate the points with high thickness, h = 700 m (even if exposed to them - if anything that would indicate underfitting). This difficulty in modeling localized high-thickness data is likely due to its underrepresentation in the training dataset (see Supplementary Information Fig. 1). While supervision helps in some cases, it does not equally enhance individual glacier outputs. Overall, we can say that the model is able to assimilate data when it becomes available (reinforcing the advantages of a machine learning approach), but where additional data should be collected for an improved prediction is non trivial.

[Figure]

Figure 1: Modeled Kongsbreen glacier (RGI60-07.01482, Svalbard), with added supervision. Big circles are ground truth data.

**L161** Shouldn't this appear in some kind of 'data availability' statement? Yes, moved.

**L209** Malaspina is currently land terminating, so its terminus is mostly grounded at or above sea level. When using 'terminus' do you mean the piedmont lobe? Yes, we replaced terminus with piedmont lobe. The recent campaign by Tober et al. (2023) has shown that the piedmont lobe is significantly grounded below sea level (See their Fig. 3.).

**L227** I am not sure how this argument is justified. Is it possible to be more quantitative? We quantify the error reduction with-supervision for both the Malaspina and Mitte glaciers, with factors of 4.7 and 2.5, respectively, highlighting that providing tie-points for the model does not have the same effect. We provide a possible explanation for this difference (providing high-thickness values is likely more important). The sentence "adding supervision does not change the solution substantially" is vague, and we removed it.

**L234** 'Conversely, the features not based on satellite products are not discrete'. I don't understand this sentence. Thanks. There was a "not" too much. Also, the sentence is not clear, and we replaced it with "Other features are per-glacier constants."

**L236** There are fundamental limits to the resolution of bed features that can be determined solely from surface observations due to the diffusive nature of ice flow. Thanks for the comment. We now better specify that the details we refer to are those of the modeled ice thickness.

**L267** These conclusions are fine, but please write them in narrative form, rather than as bullet points. The conclusions are now presented in a narrative form.

**Referee no. 2, December 9, 2024**

 The paper presents a machine learning framework to model individual glacier thicknesses based on global glacier inventory data, supplemented with other data. The paper is motivated by differences in estimated ice volume between current physics-based models. The results obtained in the paper are promising and appears to improve the ice thickness estimates compared to the existing models. Estimating the World's total glacier volume is relevant to projections of ice mass loss and sea level rise, and making use of machine learning to address this task is interesting and timely. We thanks Referee no. 2 for taking the time to review our manuscript. Please find our answers in blue.

 However, while the results are promising, the presentation needs to be improved. The **method is not sufficiently explained** but it is clear that it builds on knowledge from previous work and physics-based knowledge. The **results needs to be systematically assessed** and **limitations discussed** in more detail. Including these comments and arguments will increase confidence in the results and improve the impact. These points are explained below. We have now provided a significant amount of extra details and material to better explain the model, and the features. Please see our individual replies below. I have several concerns with the paper:

1. The selection of **features** used to train the IceBoost are not substantiated and explained. The datasets used to produce the features are from well know data repositories or data products, and clearly referenced, but it is not explained how the features in Table 1 were chosen. For example, how are the three aspect features defined, and what is the basis for the selected smoothed slopes? Were other smoothings and other variables investigated, but not included in the analysis in the end? In this version of the manuscript, the features are explained more clearly (Sect. 2.2, pages 4-8). The local Aspect feature has been dropped, and we now use only a glacier-mean Aspect value calculated directly from the DEM, eliminating the need for imputation. The use of different kernels to smooth slope, surface velocity, and curvature allows the model to capture features across multiple spatial scales (see also our response to Reviewer 1, page 3). For instance, small kernels are more beneficial for small alpine glaciers, while larger kernels are more effective for capturing changes over longer distances in extensive polar ice caps. This behavior is confirmed by the SHAP analysis, where small kernels are prioritized for alpine regions (e.g., RGI-11, European Alps, Fig. 2, right).

    The idea of using different kernels is also discussed in Millan et al. (2022), where the authors use a single adaptive kernel that progressively enlarges based on the distance from the glacier border. In contrast, our machine learning approach incorporates multiple features calculated with different kernels, allowing the model to decide how to use them when splitting the data to construct the decision tree solver. No additional kernels were investigated beyond those specified. The features excluded in this version include local Aspect, Terminus Type, and Form. The SHAP analysis was instrumental in evaluating the predictive power of all features, guiding the decisions on whether to drop or retain them, alongside considerations of computational cost and reliance on external databases (e.g., RGI), which are mutable over time. The model now relies on the RGI only for glacier geometries and their connectivity.

2. Furthermore, the main text does not define or describe the features, but the reader is referred to Appendix A and B. This makes the presentation of the model to appear rather superficial and makes it impossible to read Table 1, as well as understanding the discussions later. I suggest that some of Appendix A and B is included in the main text, so the presentation of the model appears clear and self-contained. We agree. We now

reserve a separate section in the main text for the features (Sect. 2.2, pages 4-8). Only the local mass balance feature is discussed in the Appendix, as it is rather long.

3. This brings me to the next point, which is that the physical basis for the selected features is not described or commented on. I acknowledge that the machine learning method is not based on physical relationships, but the selection of the features are building on knowledge of physics-based relationships with ice thickness and volume. I miss some comments about the relevance of these features. Some may be obvious, like slope, distance to margin, etc., but others are not obvious, and other obvious parameters are not included, like e.g. precipitation. In this context, I also miss some references to previous works of what are the key parameters, in addition to the references to the textbook and the review of scaling relationships by Bahr et al. For example work by Oerlemans (e.g. summarized very briefly in Oerlemans (2005): Extracting a climate signal from 169 glacier records. Science 308, 675-677, https://www.science.org/doi/10.1126/science.1107046 or more elaborated in Oerlamans's online booklet. In the features section, we outline the motivations behind the feature selection, add references, and elaborate further in the individual subsections. Notably, the model uses two mass balance information: a glacier-average feature from Hugonnet et al. (2021) and a (crude) downscaled map, rather than relying solely on precipitation or runoff. See also SP2.

4. It is later discussed in section 3 that some features are not important for the results, while other features are. In this discussion, it would be helpful to include an assessment of how this analysis compares to the previous work, i.e. whether the same features come out as important, or other parameters are more important in the work here. It is for example expected that slope is important. It is not a surprise that ice velocity is less important, but it is indeed very interesting to have the importance of including velocity quantified. **This should be clarified and assessed in relation to previous knowledge.** I also miss some discussion of why velocity is only important in high-latitude regions, not elsewhere. The reasoning behind why velocity is more informative at high latitudes has been clarified. We suspect that at lower latitudes, the model can predict ice thickness with sufficient confidence using other geodetic variables and metrics of size. Adding velocity, while still valuable (as indicated by the power of a SIA model), provides limited additional information in our setup. In contrast, at polar latitudes, some variables lose significance. For instance, local slope is less informative for extensive and relatively flat Arctic ice caps, and curvature is similarly less relevant. As a result, ice velocity becomes one of the few remaining local features of significance, which may explain its importance in these regions.

5. The assessment of the model results, including comparison with other results and discussion of any limitations/difficulties appears sporadic. Perhaps because it is distributed over several sections (2.4, table 2, section 4). The quantitative assessment is summarized in Table 2, but only referred to in the methods section, and the application section is very qualitatively focused on the examples. The discussion in section 4 is very interesting, but could perhaps be summarized into a more general discussion of limitations/advantages of the method. It would be very helpful/interesting to see examples where the model do not perform well. Overall, I recommend that the assessment of the results and discussion of limitations should be more systematically organized to become more convincing. Section 2.6 (and Table 2) describes the model performance when ground truth data is available. We have now expanded this section by including additional figures (Supp. Info., Figs. 3,4). We have also introduced a comparison between XGBooost and CatBoost (Supp. Info., Table 1 and Fig. 2). We have included a discussion of the model limitations in the

main text, Sect. 5, page 19 "Applications, improvements, and limitations", with figures where the model shows artifacts (Appendix, Fig. B3, page 29).

6. Another problem in this context is that the figure captions are generally deficient, in particular figures 3, 4, 5, and it took me very long time to understand what they actually show (see below). The images and captions have been improved.

Specific points:

**SP1** Section 2.4, lines 146-149: Explain why some regions are similar, so evaluations can be indicative in these regions. We rephrased this paragraph to clarify (see also comment from Reviewer1). *"The model performance is reported in Table 2 for regions with sufficient data in the training set (Fig. 1). We cannot evaluate the model performance in regions 2, 6, 9, 14, 15, 16, and 18 due to too few or absent data. For regions 10, 12, 13, and 17, where the limited data is available, statistics are provided but considered only indicative of regional performance. Nevertheless, a similar model behavior is likely expected for regions that are geographically close, have a similar ice flow regime and similar mean thickness or feature values: 13-14-15 (extensive Himalayan glaciers), and 6-7-9 (high latitude glaciers and ice caps), and 8-11-12-18 (small-to-medium size mountain glaciers)."*

**SP2** Appendix B2: I don't think it is clear why missing glacier aspect data are substituted with zero. Please explain. The local Aspect feature is not used anymore, as it adds computational burden and not particularly informative. The mean glacier aspect feature is directly calculated using the DEM, which is complete and no imputation policy is therefore necessary anymore. As a general comment, imputation policies are delicate for important features, namely velocity, where gaps are often present in the products we use. We tried as much as possible to impute by filling the gaps with meaningful values. A substantial amount of time was spent in dealing with velocity gaps and on the choice of library to deal with them. We found that Astropy has a smarter way to interpolate Nans compared to existing libraries, see `https://docs.astropy.org/en/stable/convolution/index.html`. The choice of the features to inject into the model was also driven by any gaps in the product: with a global model we had to be sure that all features are available for all glaciers, and if not, that the imputation policy was robust enough, as the model cannot accept a NaN as input. For example, we investigated adding the RUNOFF variable from MERRA-2. The problem with such a feature is that the MERRA-2 glacier masks do not reflect RGI glacier outlines, and, as a result, a void filling pipeline was attempted to estimate RUNOFF for those glaciers not covered by MERRA-2. Such pipeline ended up not robust enough, and this feature was discarded.

**SP3** Figure captions: all figure captions need to be reviewed and checked for completeness and clarity. Add letters a, b, etc The captions have been improved and the letters on the figures have been added.

**SP4** Figure 3 and 4: it was not clear that the two models in lower row show the same results as in the upper row, but now with the observations added. Please clarify that observations are added in all three figure in lower row. This will help underline how much better the IceBoost is in these examples. We now simplified these figures, by only showing the Millan and Farinotti models with data added. These 2 models with and without data do not change so there were 2 useless panels now removed.

**SP5** Figure 5: It is not clear what big and small circles mean. Only by zooming in, it is possible to distinguish. It is not explained what the background image is. We now better

clarify that the small circles reflect the model solution (the arbitrary/random locations where the model is queried), while the big circles reflect the ground truth data. We added an explanation regarding the DEM hillshade background in all figures where the hillshade is shown.

**SP6** I suggest to merge section 4 and 4.1, as section 4.1 adds to the impression that the assessment in the main paper is lacking. The discussion is now presented in a unique section (Sect. 4). This section has also been expanded with a SHAP analysis, to investigate the feature rankings on an individual-glacier basis. In relation to the model discussion, we added a separate Sect. 5 "Applications, improvements, and limitations".

**Comments from Editor Ludovic Räss**

- You may want to use semantic versioning to allow including minor and patch releases in the future for your software, thus modifying your title to v1.0.0. Thanks. We will use the major.minor versioning - the new title has been changed accordingly.

- Extend the content about reporting on the model's performance, expanding on how the results differ from existing approaches. We have added some text in the introduction regarding the novelties of our approach, as compared to previous ones. Regarding the model's performance (where ground truth allows to evaluate it), we added in the Supplementary Information, 2 representative figures that support the how the performance of the model is evaluated (Table 2). This analysis is done for in region 3 (Arctic Canada N.) as a case study.

- It would be helpful and informative to have more supporting figures instead of tables when it comes to comparison and evaluation of model output, in order to better support the improvement compared to existing models. We have now included relevant figures in a Supplementary Information file. We also include in that Supp. Info the n=190 comparative figures (Iceboost, Millan et al. 2022 and Farinotti et al. 2019) that were previously deposited on Zenodo. Although a rather large file (30 MB), we believe this is a convenient way to group all material in the relevant place.

---

## Author Response (AR2)

**1 Topic editor by Ludovic Räss, 17 Feb 2025**

Dear authors,

Thank you for providing a revised version of your manuscript draft. As supported by the external reviewer, your current manuscript version significantly improved. As remaining step, I would like you to address the following item, as both the external reviewer and myself would like you to include further details about this in the text. Namely, it would be valuable to better describe how dependent your results are on processed data from other models (e.g. mass balance, MB and mb). As the external reviewer suggests, "these derived data have uncertainties and limitations, and it would be interesting to hear how dependent the model are on these data. The Shap analysis provides some input to this, but some remarks on this would be interesting."

Thank you for your work, best regards Ludovic Räss

**2 Report by Anonymous referee #2, 16 Feb 2025**

Overall, I find that the revised manuscript is significantly improved. It contains now a more detailed motivation and perspectives to previous work, the methods and selection of datasets are better described and well-argued, and the presentation of the model appears now self-contained. The revised structure and including more explanation in the main manuscript has contributed to this, as well as more elaborated explanations. The description of datasets and why these are chosen are now sufficiently explained, and the assessment of the results in relation to these datasets is improved. Further, I find that the revised paper presents the method and results sufficiently clear. The examples presented in figures 3 and 4 are impressive and promising. My remaining considerations are now related to how dependent the results are on processed data from other models, like the mass balance, MB and mb. These derived data have uncertainties and limitations, and it would be interesting to hear how dependent the model are on these data. The Shap analysis provides some input to this, but some remarks on this would be interesting. Minor edits:

- Figure 1 caption: region 18 has also no training data. Thanks for spotting this.

- Line 388: ex-novo is not commonly used, please clarify. Changed to: "Ice velocity maps (or mass balance) can be generated from scratch or used to fill gaps in existing products."

We thank the Anonymous referee #2 for the time to review again the manuscript. We are glad that we could improve the quality of the work by making use of their comments and insights. With respect to the last comment, we here provide an answer to the Editor as well, who shared the same concern.

We agree with the Referee #2 - the Shapely analysis provides an indication on variable ranking but does not provide information on why that variable is important or not. If a variable is ranked low, it may indicate either that the variable is not as informative as the others (in the tree structure), or that the variable is extracted from a product of low(er) quality.

The DEM is by far the most important input product for the model. It generates 24 out of 39 features, some of which are very important (namely, slopes). IceBoost was originally developed on the Aster GDEM (same horizontal resolution), and the model accuracy was found significantly lower as to Tandem-X EDEM, confirming that the quality of the DEM is crucial.

The velocity product drives 6 features. The Shapely analysis shows some importance for ice velocity, but probably not as high as expected. One of the reason could be that the dataset is imperfect. The geodetic mass balance dataset is itself of 'lower quality' because it contains glacier-averaged data. The downscaled variable, mb, is found slightly more informative. We

note that the downscaling method introduced for this feature is only applied outside polar regions. In the polar regions, RACMO is used.

We have now added two appendix sections, C and D, where we investigatge the effect of the MB and mb features on the model output.

- In Appendix C, we assess the volume of the Unteraargletscher by varying the mass balance (MB) from -4 to 4 $m\ w.e\ yr^{-1}$ including the reference value of -1.59 $m\ w.e\ yr^{-1}$. The modeled volume show little overall variations, even with an unrealistically wide range of MB values. This aligns with the Shapley analysis, confirming that MB is a weak predictor, and that the results are rather insensitive to this variable.

- In Appendix D, we perform a sensitivity test on the two regional parameters used to calculate mass balance maps: $\bar{s}$ and $\bar{q}$. Using a Monte Carlo approach with 1,500 model runs and Gaussian noise added, we monitor the modeled volumes of the Unteraargletscher and Aletsch glaciers (Central Europe). We find that a 50% uncertainty in both parameters leads to an upper-limit error of approximately 15% in the modeled volumes.